# Keratins and plakin family cytolinker proteins control the length of epithelial microridge protrusions

Yasuko Inaba*, Vasudha Chauhan, Aaron Paul van Loon, Lamia Saiyara Choudhury, Alvaro Sagasti*

Molecular, Cell and Developmental Biology Department and Molecular Biology Institute, University of California, Los Angeles, Los Angeles, United States

**Abstract** Actin filaments and microtubules create diverse cellular protrusions, but intermediate filaments, the strongest and most stable cytoskeletal elements, are not known to directly participate in the formation of protrusions. Here we show that keratin intermediate filaments directly regulate the morphogenesis of microridges, elongated protrusions arranged in elaborate maze-like patterns on the surface of mucosal epithelial cells. We found that microridges on zebrafish skin cells contained both actin and keratin filaments. Keratin filaments stabilized microridges, and overexpressing keratins lengthened them. Envoplakin and periplakin, plakin family cytolinkers that bind F-actin and keratins, localized to microridges, and were required for their morphogenesis. Strikingly, plakin protein levels directly dictate microridge length. An actin-binding domain of periplakin was required to initiate microridge morphogenesis, whereas periplakin-keratin binding was required to elongate microridges. These findings separate microridge morphogenesis into distinct steps, expand our understanding of intermediate filament functions, and identify microridges as protrusions that integrate actin and intermediate filaments.

*For correspondence:
yasuko.nam@gmail.com (YI);
sagasti@mcdb.ucla.edu (AS)

Competing interests: The authors declare that no competing interests exist.

## Introduction

Cytoskeletal filaments are scaffolds for membrane protrusions that create a vast diversity of cell shapes. The three major classes of cytoskeletal elements—microtubules, actin filaments, and intermediate filaments (IFs)—each have distinct mechanical and biochemical properties and associate with different regulatory proteins, suiting them to different functions. Actin filaments create a wide variety of well-studied protrusions, including filopodia, lamellipodia, invadopodia, and microvilli (*Blanchoin et al., 2014*; *Pollard and Cooper, 2009*). Similarly, microtubules form cilia and flagella (*Mirvis et al., 2018*). By contrast, IFs are not commonly believed to directly participate in the formation of cellular protrusions.

IFs are diverse, including nuclear lamins, neurofilaments, glial fibrillary acidic proteins, vimentins, and keratins (*Etienne-Manneville, 2018*; *Herrmann and Aebi, 2016*). Although each IF type is biochemically distinct, they all share structural properties. Whereas actin filaments and microtubules lengthen by preferentially adding filaments to one end, IFs are unpolarized. Tetrameric IF subunits incorporate not only at filament ends, but also within filaments, a process called 'intercalary exchange' that allows IFs to replace subunits without altering filament structure (*Colakoğlu and Brown, 2009*). The viscoelastic properties of IFs make them the strongest of the cytoskeletal elements. IFs deform at low strains, but, whereas actin filaments and microtubules break at high strains, IFs rigidify and resist breakage (*Janmey et al., 1991*). Their stability and strength together make IFs ideal for maintaining cellular integrity.

Keratins are the most abundant IFs in epithelial cells. They organize and reinforce epithelial tissues by anchoring cells to one another and to the extracellular matrix at desmosomes and hemi-

**eLife digest** Cells adopt a wide array of irregular and bumpy shapes, which are scaffolded by an internal structure called the cytoskeleton. This network of filaments can deform the cell membrane the way tent poles frame a canvas. Cells contain three types of cytoskeleton elements (actin filaments, intermediate filaments, and microtubules), each with unique chemical and mechanical properties.

One of the main roles of the cytoskeleton is to create protrusions, a range of structures that 'stick out' of a cell to allow movement and interactions with the environment. Both actin filaments and microtubules help form protrusions, but the role of intermediate filaments remains unclear.

Microridges are a type of protrusion found on cells covered by mucus, for instance on the surface of the eye, inside the mouth, or on fish skin. These small bumps are organised on the membrane of a cell in fingerprint-like arrangements. Scientists know that actin networks are necessary for microridges to form; yet, many structures supported by actin filaments are not stable over time, suggesting that another component of the cytoskeleton might be lending support.

Intermediate filaments are the strongest, most stable type of cytoskeleton element, and they can connect to actin filaments via linker proteins. However, research has yet to show that this kind of cooperation happens in any membrane protrusion.

Here, Inaba et al. used high-resolution microscopy to monitor microridge development in the skin of live fish. In particular, they focused on a type of intermediate filaments known as keratin filaments. This revealed that, inside microridges, the keratin and actin networks form alongside each other, with linker proteins called Envoplakin and Periplakin connecting the two structures together.

Genetic experiments revealed that Envoplakin and Periplakin must attach to actin for microridges to start forming. However, the two proteins bind to keratin for protrusions to grow. This work therefore highlights how intermediate filaments and linker proteins contribute to the formation of these structures.

Many tissues must be covered in mucus to remain moist and healthy. As microridges likely contribute to mucus retention, the findings by Inaba et al. may help to better understand how disorders linked to problems in mucus emerge.

desmosomes (*Osmani and Labouesse, 2015*), and they are bundled and cross-linked during the process of keratinization to create the cornified outer layers of mammalian skin (*Eckhart et al., 2013*; *Smack et al., 1994*). Although keratins are not known to be directly involved in the morphogenesis of protrusions, they support microvilli as part of the terminal web at their base, where they interact with F-actin rootlets extending from microvilli, as well as myosin and other actin-binding proteins (*Hirokawa et al., 1982*). Keratins, along with the IF vimentin, have also been detected in long invadopodia (*Schoumacher et al., 2010*). Although keratins themselves are not required for invadopodia morphogenesis, disrupting vimentin prevents their full lengthening, suggesting that IFs may play a role in the late stages of invadopodia extension (*Schoumacher et al., 2010*).

Cytolinker proteins bind to multiple cytoskeletal elements to integrate them into cellular structures. For example, keratins are connected to F-actin in the terminal web of intestinal microvilli, potentially by the cytolinker plastin 1 (*Grimm-Günter et al., 2009*). Another family of cytolinkers, the plakins, consist of several large, multi-domain proteins that link cytoskeletal elements to cell junctions or to one another (*Jefferson et al., 2004*; *Sonnenberg and Liem, 2007*). The plakin family members periplakin (Ppl) and envoplakin (Evpl), which dimerize with each other and form hetero-oligomeric complexes (*Kalinin et al., 2004*), localize to desmosomes (*DiColandrea et al., 2000*) and are components of the cornified envelope in mammalian skin (*Ruhrberg et al., 1997*; *Ruhrberg et al., 1996*). *ppl* and *evpl* knockout mice are viable (*Aho et al., 2004*; *Määttä et al., 2001*), though skin barrier formation is delayed in *evpl* mutants (*Määttä et al., 2001*). Evpl and Ppl have large N-terminal regions with direct actin-binding activity (*Kalinin et al., 2005*), as well as domains that associate with actin-binding proteins in other plakin family members (*Jefferson et al., 2004*; *Sonnenberg and Liem, 2007*). Plakins also have rod domains that form coiled-coils mediating dimerization (*Kalinin et al., 2004*), and C-terminal domains that bind to IFs (*Karashima and Watt,*

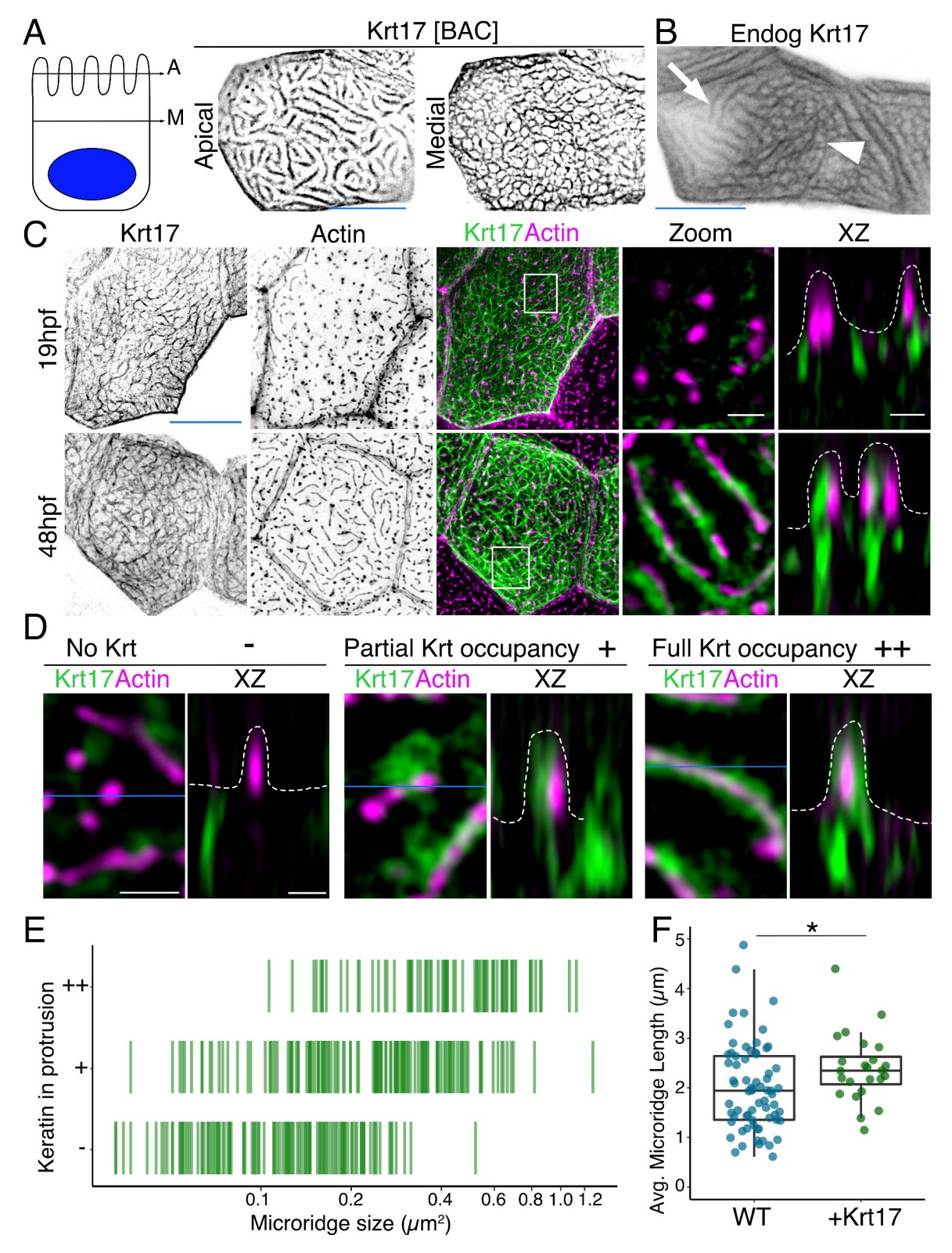

**Figure 1.** Keratins are core components of microridges. (**A**) SIM optical sections of a Krt17-GFP[BAC]-expressing zebrafish periderm cell at 48hpf. Cartoon shows the relative location of apical (A) and medial (M) optical sections. (**B**) Oblique optical section through a cell with endogenously-tagged Krt17. This section shows both the microridge-like pattern at the apical cell surface (arrow) and a filamentous pattern deeper in the cell (arrowhead). (**C**) Projections and orthogonal views of SIM images of Krt17-GFP[BAC]- and Lifeact-mRuby-expressing cells at the indicated developmental stages. White

*Figure 1 continued on next page*

Figure 1 continued

boxes, regions of magnification in zoom panels. Orthogonal views (XZ) at 19hpf show that Krt17 is not in pegs but at 48hpf localizes alongside F-actin in microridges. Dashed lines outline cell membranes. (D) Section (XY) and orthogonal (XZ) views of SIM images of Krt17-GFP[BAC]- and Lifeact-mRuby-expressing cells at 48hpf. '-': No keratin in protrusion. '+': Keratin partially occupies microridge. '++': Keratin fully occupies microridge. Blue lines show the location of XZ optical sections. Dashed lines outline cell membranes. (E) Plot showing keratin localization in actin protrusions of different sizes (area) at 48hpf. The presence of keratin was scored as in D. n = 3 cells from two fish. (F) Dot and box-and-whisker plots of average microridge length per cell in WT cells expressing Lifeact-mRuby, and cells expressing Krt17-GFP[BAC] and Lifeact-mRuby at 48hpf. *p<0.05, the Wilcoxon rank-sum test. n = 24 cells from four fish in Krt17-GFP[BAC]-expressing fish and 69 cells from nine fish in WT. WT data is the same as in *Figure 4B*. Box-and-whisker plots (F): Middle line shows the median; the upper and lower ends of the box are the 75th and 25th percentiles. Black-and-white images were inverted so that high-intensity fluorescence appears black and low fluorescence is white. Scale bars: 10 μm (A-C, blue line) and 1 μm (C-D white line).

The online version of this article includes the following figure supplement(s) for figure 1:

**Figure supplement 1.** Keratins localize in two patterns in periderm cells.

*2002*; *Kazerounian et al., 2002*). Thus, Evpl and Ppl have the potential to link F-actin with keratin filaments.

In this study, we investigated the relationship between keratins, F-actin, and plakins in the morphogenesis of microridges, which are laterally elongated protrusions arranged in maze-like patterns on the apical surface of epithelial cells (*Depasquale, 2018*). Microridges are formed on a variety of mucosal epithelial cells, including cells that make up the outer layer of the zebrafish epidermis, called the periderm, where they are required to maintain glycans on the skin surface (*Pinto et al., 2019*). Microridge protrusions are filled with F-actin but are more persistent than several better studied actin-based structures, such as lamellipodia and filopodia. Microridges are formed from the coalescence of finger-like, actin-based precursor protrusions called pegs (*van Loon et al., 2020*), a process dependent on the F-actin nucleator Arp2/3 (*Lam et al., 2015*; *Pinto et al., 2019*; *van Loon et al., 2020*) and the relaxation of surface tension by cortical myosin-based contraction (*van Loon et al., 2020*). Although studies of microridge morphogenesis have exclusively focused on F-actin regulation, like microvilli, microridges have keratins at their base, and ultrastructural studies have reported the occasional presence of IFs within microridges (*Pinto et al., 2019*; *Schliwa, 1975*; *Uehara et al., 1991*).

Using a combination of live imaging, mutant analysis, and structure-function studies, we found that keratins are integral components of microridges, and that Evpl and Ppl control microridge stability and length by recruiting keratin cytoskeletal filaments. Thus, F-actin-keratin cytolinkers create a hybrid cytoskeletal scaffold that enables the morphogenesis of microridge protrusions.

## Results

### Keratins are core components of mature microridges

To investigate keratin localization in the zebrafish periderm, we tagged six type I keratin proteins expressed in periderm cells (*Cokus et al., 2019*) with GFP or mRuby at their C-termini, using bacterial artificial chromosomes (BACs). Imaging periderm cells in live zebrafish expressing these reporters revealed that all keratins localized in two distinct patterns within a cell: As expected, they formed a filamentous network filling cells; remarkably, they also formed what appeared to be thick bundles in the pattern of microridges at the apical surface (*Figure 1A*, *Figure 1—figure*

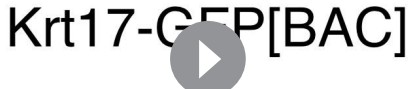

**Video 1.** Keratins localize in microridge-like patterns apically and to filamentous networks throughout periderm cells. Apical-to-basal SIM (Krt17-GFP[BAC]) and confocal (others) optical sections of periderm cells expressing six different fluorescently-tagged Type 1 keratins, as indicated by title cards.
https://elifesciences.org/articles/58149#video1

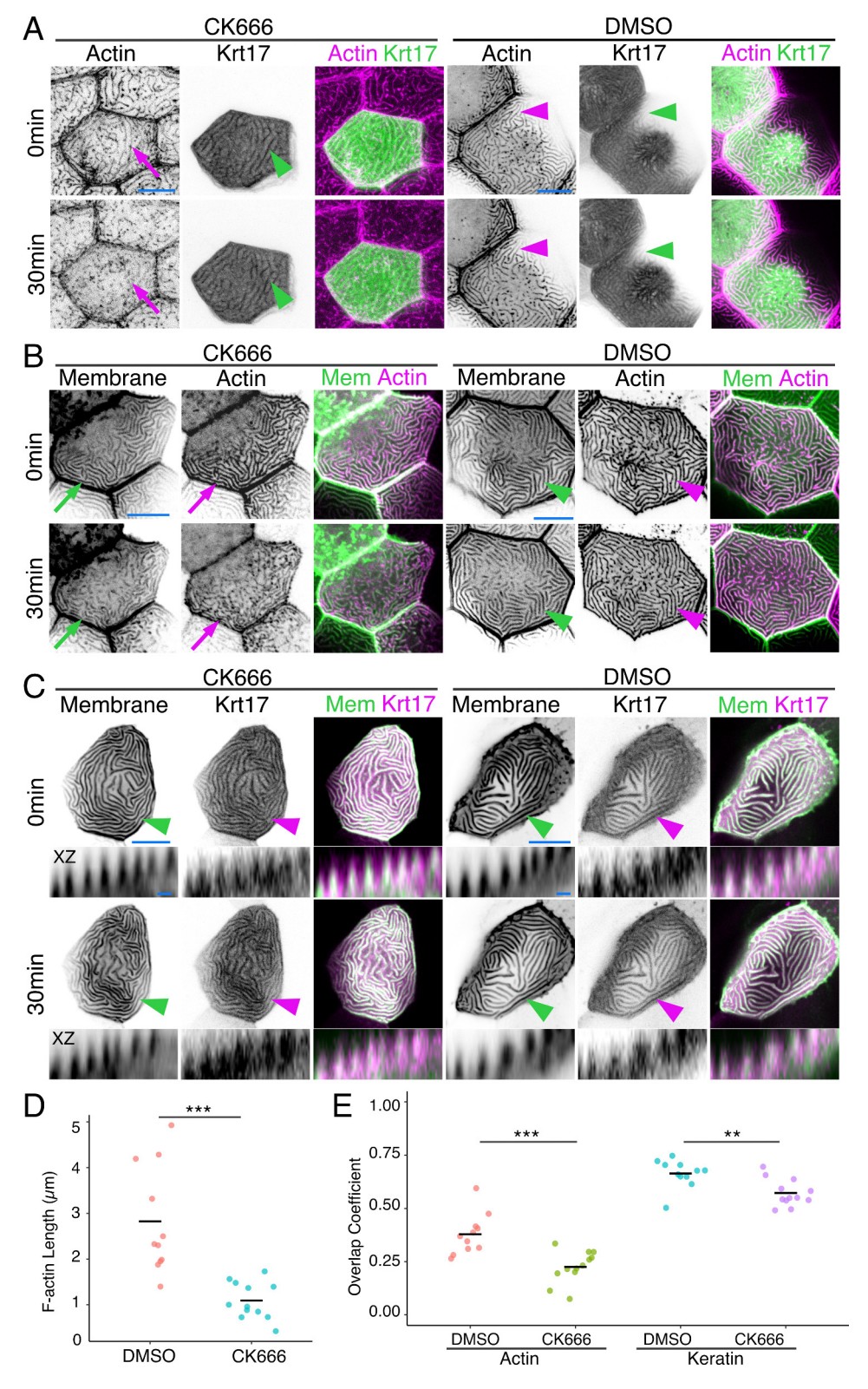

**Figure 2.** Keratins retain microridge structure. (**A**) Krt17-GFP[BAC]- and Lifeact-mRuby-expressing cells in 48hpf fish, at time 0 and 30 min after treatment with DMSO or the Arp2/3 inhibitor CK666. Arrows show that the F-actin microridge pattern was disrupted after 30 min of CK666 treatment. Arrowheads show that the keratin microridge pattern was retained after 30 min of CK666 or DMSO treatment. (**B**) GFP-PH-PLC (membrane) and Lifeact-mRuby in 48hpf periderm cells at time 0 and 30 min after treatment with DMSO or CK666. GFP-PH-PLC was expressed by injecting Krt5:Gal4 into UAS:
*Figure 2 continued on next page*

*Figure 2 continued*

GFP-PH-PLC fish; Krt5:Lifeact-mRuby was expressed by transient transgenesis. Arrows show that the membrane and actin microridge patterns were disrupted after 30 min of CK666 treatment. Arrowheads show that the membrane and actin microridge patterns were retained in DMSO controls. (C) Projection and orthogonal views of GFP-PH-PLC and Krt17-mRuby[BAC] in 48hpf periderm cells at time 0 and 30 min after treatment with DMSO or CK666. GFP-PH-PLC was expressed by injecting Krt5:Gal4 into UAS:GFP-PH-PLC fish; Krt17-mRuby[BAC] was expressed by transient transgenesis. Arrowheads show that membrane protrusion and Krt17 microridge patterns were retained after 30 min of DMSO or CK666 treatment. Orthogonal views (XZ) show that Krt17 preserved the protrusive membrane structure after 30 min of CK666 treatment. (D) Dot plot of average microridge length per cell at 48hpf after 30 min treatment with DMSO or CK666. Line indicates average. ***p<0.005, the Wilcoxon rank-sum test. n = 11–12 cells from three fish. (E) Dot plot of microridge and keratin overlap coefficients, comparing the colocalization of each protein at 0 min with its localization after 30 min DMSO or CK666 treatment. Line indicates average. **p<0.01, ***p<0.005, the Wilcoxon rank-sum test. n = 11–12 cells from three fish. Black-and-white images were inverted so that high-intensity fluorescence appears black and low-intensity fluorescence is white. Scale bars: 10 µm (A–C) and 1 µm (orthogonal images in C).

*supplement 1*, *Video 1*). Tagging an allele of one of these keratins, keratin 17 (Krt17), with CRISPR-facilitated homologous recombination, confirmed that the endogenously expressed protein localized in these two patterns (*Figure 1B*).

To observe keratin localization at higher resolution, we used super-resolution structured illumination microscopy (SIM) to image cells expressing the Krt17-GFP BAC reporter. At an early stage of microridge morphogenesis (19 hr post-fertilization, hpf), when periderm surfaces are dominated by pegs (*van Loon et al., 2020*), Krt17 formed the filamentous pattern in cells but did not localize within pegs. At a later stage (48hpf), when mature microridges have formed, Krt17 invaded microridges, where it appeared to form filaments alongside F-actin (*Figure 1C*). Scoring the presence of keratin in protrusions of different lengths confirmed that the smallest protrusions, pegs, are largely devoid of keratin, and that keratin localization to microridges increased as they lengthened (*Figure 1D–E*). Intriguingly, the average microridge length per cell, as measured by imaging the actin reporter Lifeact-mRuby, was slightly longer in Krt17-GFP over-expressing cells than in wild-type (WT) cells (*Figure 1F*). These observations confirm that keratins are components of mature microridges and suggest that they may play a role at later stages of microridge morphogenesis.

## Keratins are stable microridge components

Since IFs are the strongest and most stable of the cytoskeletal filaments, we wondered if they might contribute to microridge stability. To destabilize F-actin in microridges, we treated animals for 30 min with the Arp2/3 inhibitor CK666, which causes the F-actin in microridges to disassemble and redistribute back into peg-like structures (*Figure 2A and D*; *Lam et al., 2015*; *Pinto et al., 2019*; *van Loon et al., 2020*). Strikingly, despite the loss of the F-actin microridge pattern, the apical Krt17-GFP microridge pattern was retained (*Figure 2A*; *Figure 2E*). Labeling cells with a membrane reporter (GFP-PH-PLC) revealed that overexpressing the Krt17-GFP BAC reporter preserved the protrusive membrane topology upon F-actin disruption (*Figure 2B*; *Figure 2C*). These results suggest that keratins may play roles in stabilizing and/or elongating microridges at later stages of morphogenesis.

## The cytolinker proteins envoplakin and periplakin localize to microridges

If F-actin and keratins both contribute to microridge morphogenesis, we speculated that they may interact via linker proteins. By examining larval periderm cell transcriptomes (*Cokus et al., 2019*), we identified two potential cytolinker proteins, Evpl and Ppl, which are highly expressed and enriched in periderm cells, relative to other epithelial cells. Evpl and Ppl, members of the plakin protein family (*Jefferson et al., 2004*), contribute to keratinization of the mammalian skin (*Ruhrberg et al., 1997*; *Ruhrberg et al., 1996*), heterodimerize with each other (*Kalinin et al., 2004*), and can bind both F-actin (*Kalinin et al., 2005*) and keratins (*Karashima and Watt, 2002*; *Kazerounian et al., 2002*), thus potentially linking the two types of cytoskeletal filaments in microridges. To determine the localization of Evpl and Ppl in zebrafish periderm cells, we made Ppl-GFP, Evpl-GFP, and Evpl-mRuby BAC reporter fusions and imaged them in transient transgenic animals. These reporters were expressed in periderm cells throughout the animal and, as expected from genomic analyses (*Cokus et al., 2019*; *Liu et al., 2020*), were apparently exclusive to periderm cells. Evpl and Ppl both

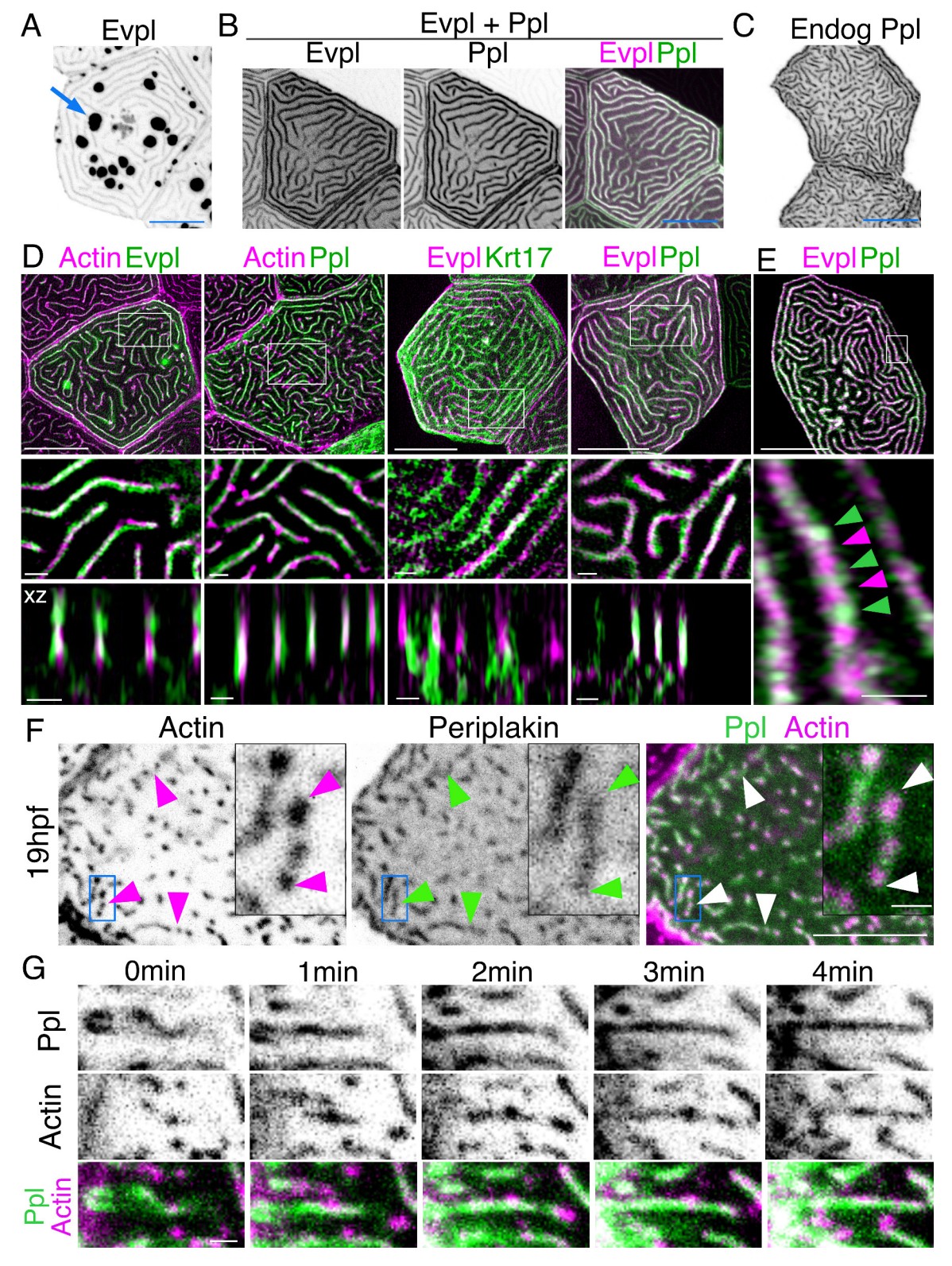

**Figure 3.** Evpl and Ppl localize to microridges and are required for initiating microridge morphogenesis. (A) Evpl-mRuby[BAC]-expressing periderm cells in 48hpf zebrafish show Evpl localization in aggregates (e.g. arrow) and microridges. (B) Evpl-mRuby[BAC] and Ppl-GFP[BAC] co-expression in a periderm cell at 48hpf. (C) Endogenously GFP-tagged Ppl in periderm cells at 48hpf. (D) Projection and orthogonal views of SIM images of the indicated co-expressed reporters. White boxes, regions of magnification in middle panels. Bottom panel, orthogonal view (apical up, basal down). (E)

*Figure 3 continued on next page*

*Figure 3 continued*

Confocal Airyscan image of cells expressing Ppl-GFP[BAC] and Evpl-mRuby[BAC]. White box, region of magnification in lower panel. Arrowheads show alternating arrangement of Ppl and Evpl in microridges. (F) Ppl-GFP[BAC] and Lifeact-mRuby expression in a 19hpf periderm cell. Arrowheads point to representative areas in which Ppl localizes to longer structures than actin pegs. Boxes, area of magnification for insets. (G) Sequential projections from a time-lapse movie of Ppl-GFP[BAC]- and Lifeact-mRuby-expressing periderm cells at the end of cytokinesis (24hpf). Note that Ppl structures appear to precede F-actin in developing protrusions. Black-and-white images were inverted so that high-intensity fluorescence appears black and low-intensity fluorescence is white. Scale bars: 10 µm (**A–F**) and 1 µm (zoomed and orthogonal views in D, F, and G).

The online version of this article includes the following figure supplement(s) for figure 3:

**Figure supplement 1.** Evpl associates with Ppl in microridges.

localized to microridges (*Figure 3A–E*). Similar to its behavior in mammalian cell culture (*DiColandrea et al., 2000*), when expressed on its own, Evpl formed prominent aggregates, which were reduced when Ppl was co-expressed (*Figure 3A–B*, *Figure 3—figure supplement 1*), consistent with the possibility that the two plakin proteins dimerize or oligomerize. GFP-tagging the endogenous gene with CRISPR-facilitated homologous recombination verified the Ppl localization pattern (*Figure 3C*). SIM microscopy of the co-expressed BAC reporters revealed that Evpl and Ppl were localized within microridges. In optical sections along the z-axis, Ppl and Evpl appeared to be adjacent to F-actin and keratin filaments but were almost completely overlapping with each other (*Figure 3D*). In an x-y section, however, Evpl and Ppl formed an apparently alternating pattern, consistent with their ability to assemble into a higher order oligomeric arrangement (*Kalinin et al., 2004*; *Figure 3E*).

To determine when plakins first localize to microridges, we observed Ppl localization at an earlier stage (19hpf), when periderm cell surfaces are dominated by pegs. Unlike Krt17, at this stage, Ppl associated with pegs and, surprisingly, often appeared to form longer and more continuous structures than F-actin itself (*Figure 3F*). To observe their localization with greater temporal precision, we imaged Ppl and F-actin reporters in time-lapse movies of cells undergoing cytokinesis. Just before periderm cell division, microridges dissolve, but then re-assemble at the end of cytokinesis (*Lam et al., 2015*), thus allowing us to image the entire process with rapid, predictable kinetics. These movies revealed that Ppl formed elongated, continuous structures immediately before the coalescence of F-actin pegs into microridges, potentially as part of a template for microridge assembly (*Figure 3G*, *Video 2*). Since keratins do not localize to microridges until later in development, these observations suggest that Ppl likely plays a keratin-independent role in the initiation of microridge morphogenesis.

## Envoplakin and periplakin dictate microridge length

To determine the function of Evpl and Ppl in microridge morphogenesis, we created stable zebrafish mutant lines by deleting several exons of each gene with the CRISPR/Cas9 system (*Figure 4*, *Figure 4—figure supplement 1*). At larval stages, *ppl* mutants lacked microridges, forming only pegs, whereas *evpl* mutants formed pegs and short microridges (*Figure 4A–B*, *Figure 4—figure supplement 2A–B*, *Supplementary file 1*). Periderm cells in double mutants were indistinguishable from those in *ppl* mutants, projecting only pegs. Evpl and Ppl BAC reporters rescued microridge formation in each mutant (*Figure 4—figure supplement 3*), demonstrating that the BAC fusions were functional and verifying that the mutations reduced gene function. This conclusion was further supported by the observation that morpholino antisense oligonucleotides targeting *evpl* and *ppl* caused similar microridge defects, which were rescued by expressing cDNAs of each gene (*Figure 4—figure supplement 4*). Microridge defects persisted through adulthood in both genetic mutants (*Figure 4—figure supplement*

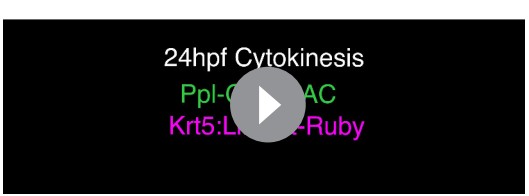

**Video 2.** Periplakin forms continuous structures during peg coalescence at the end of cytokinesis. Time-lapse confocal video of periderm cells expressing Lifeact-mRuby and Ppl-GFP[BAC] immediately after cytokinesis. A second video shows a magnified view. Time stamps in the upper left.
https://elifesciences.org/articles/58149#video2

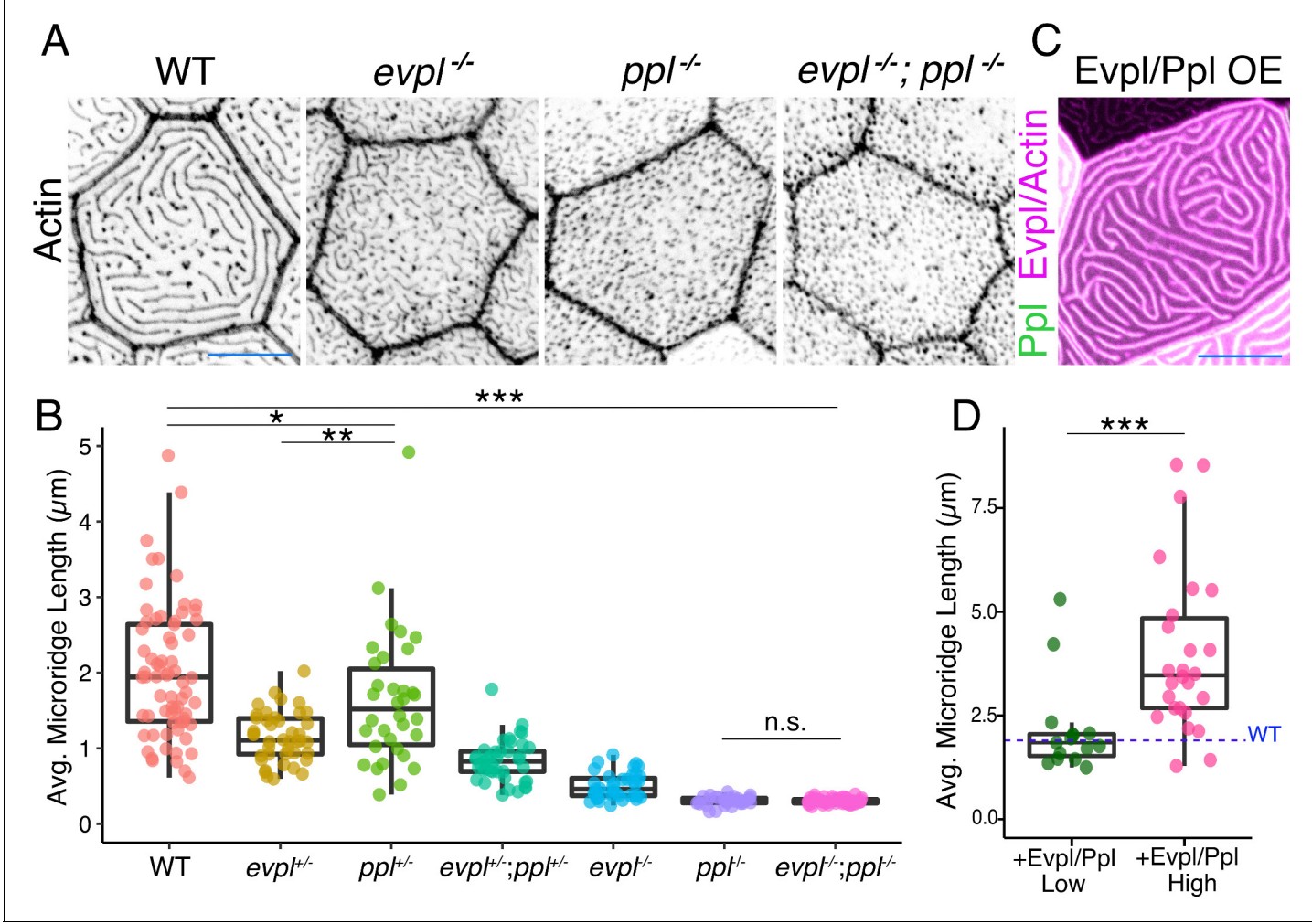

**Figure 4.** Evpl and Ppl are required for microridge morphogenesis and determine their length. (A) Periderm cells expressing Lifeact-mRuby in WT, *evpl*^−/−^, *ppl*^−/−^, and *evpl*^−/−^;*ppl*^−/−^ mutants at 48hpf. Images were inverted so that high-intensity fluorescence appears black and low-intensity fluorescence is white. (B) Dot and box-and-whisker plot of average microridge length per cell at 48hpf from animals of the indicated genotypes. All comparisons between each genotype were significantly different from one another, except where indicated as n.s. *p<0.05, **p<0.01; for all other comparisons ***p<0.0001, the Wilcoxon rank-sum test. The exact P values for all comparisons are shown in **Supplementary file 1**. n = 27–69 cells from 3 to 9 fish per genotype. Another version of this graph color-coding cells from each animal is provided in **Figure 4—figure supplement 2A**, as well as a violin plot showing the length distributions of all microridges pooled (**Figure 4—figure supplement 2B**, **Supplementary file 1**). To control for animal-to-animal variation, we have also analyzed these data by averaging the microridge length averages for each cell in each animal (i.e. each animal is represented by one number averaging all cells). This approach yielded similar results and are reported in **Supplementary file 1**. (C) Cells overexpressing Evpl-mRuby[BAC], Ppl-GFP[BAC], and Lifeact-mRuby at 48hpf. (D) Dot and box-and-whisker plot of average microridge length per cell in cells over-expressing Evpl-mRuby[BAC] and Ppl-GFP[BAC] at 48hpf. See **Figure 4—figure supplement 7** for categorization into 'low' and 'high' overexpression groups. ***p<0.0001, the Wilcoxon rank-sum test. n = 41 cells from five fish. Dotted blue line shows the median average microridge length per cell in WT animals (from B). Box-and-whisker plots (B and D): Middle line shows the median; the upper and lower ends of the box are the 75th and 25th percentiles. Scale bars: 10 μm.

The online version of this article includes the following figure supplement(s) for figure 4:

**Figure supplement 1.** *evpl* and *ppl* mutants.

**Figure supplement 2.** Additional quantification of microridge length and cell area in *evpl* and *ppl* mutants.

**Figure supplement 3.** Plakin BAC reporters rescue microridge development in plakin mutants.

**Figure supplement 4.** *evpl*- and *ppl*-targeting morpholinos disrupt microridge development.

**Figure supplement 5.** Microridges in adult fish were disrupted in *evpl* and *ppl* mutants.

**Figure supplement 6.** *evpl* and *ppl* mutant morphologies.

**Figure supplement 7.** Evpl and Ppl dictate microridge length.

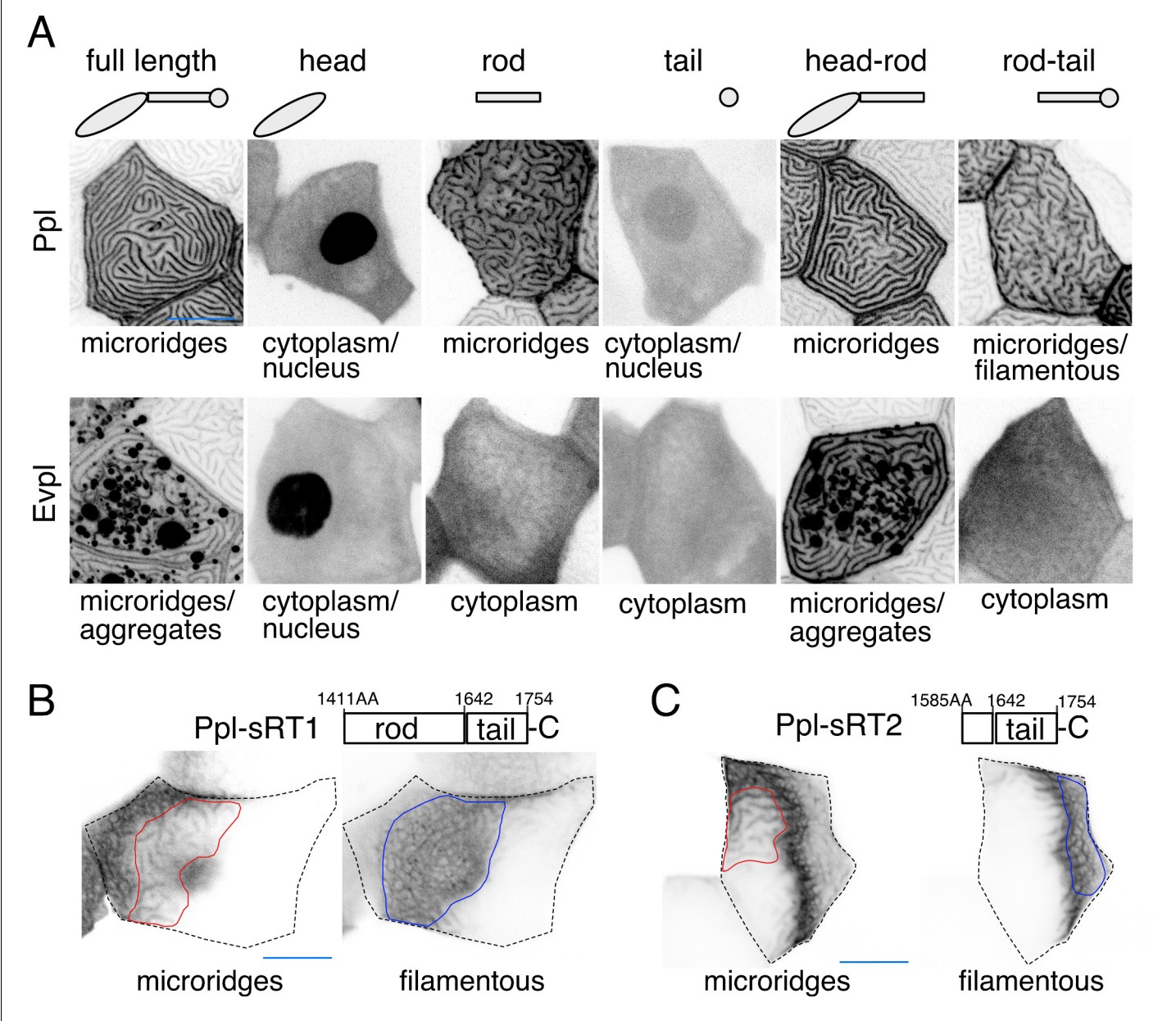

**Figure 5.** Ppl and Evpl domain localization. (**A**) Periderm cells expressing Evpl-tdTomato and Ppl-GFP variants at 48hpf. Schematics indicate the domains in each variant. (**B–C**) Optical sections of GFP-tagged Ppl truncated rod-tail fusions expressed in periderm cells at 48hpf. Sections highlight the microridge-like pattern at the apical surface of cells (left, red outlines) or the filamentous pattern deeper in cells (right, blue outlines). Top: Diagram of Ppl protein domains. Amino acid numbers are indicated. See *Figure 5—video 1* for video showing apical-to-basal sections of these images. Images were inverted so that high-intensity fluorescence appears black and low-intensity fluorescence is white. Scale bars: 10 μm.

The online version of this article includes the following video and figure supplement(s) for figure 5:

**Figure supplement 1.** Evpl depends on Ppl for apical localization, but not vice versa.

**Figure 5—video 1.** Truncated periplakin rod-tail domains localize to microridges and the IF filamentous network.

https://elifesciences.org/articles/58149#fig5video1

---

**5**) but all single and double mutant animals were homozygous viable, fertile, and appeared morphologically normal at all stages (*Figure 4—figure supplement 6*). Apical cell areas were comparable between the wildtype and mutant larvae, indicating that these mutations do not cause cells to become dysmorphic or compromise apical constriction (*Figure 4—figure supplement 2C*, *Supplementary file 1*). However, apical cell areas were, on average, significantly smaller in double

mutants, suggesting that Evl and Ppl may play an unrelated, redundant role in restraining apical constriction.

Unexpectedly, in both *evpl* and *ppl* heterozygotes, average microridge length per cell was shorter than in WT cells but longer than in homozygous mutants (*Figure 4B*, *Supplementary file 1*). Average microridge length was shorter in trans-heterozygous mutants (*evpl*$^{+/-}$; *ppl*$^{+/-}$) than in either heterozygous mutant alone but longer than in homozygous mutants. These observations reveal that *evpl* and *ppl* mutants are semi-dominant and that the dose of plakin proteins dictates microridge length.

To determine if Evpl and Ppl are not only required, but also sufficient for lengthening microridges, we co-overexpressed the Evpl-mRuby and Ppl-GFP BAC reporters in WT animals (*Figure 4C*). Since the reporters were fluorescently tagged, we estimated the relative concentration of overexpressed plakins in each cell from its fluorescence intensity. Plotting fluorescence intensity versus average microridge length per cell revealed that cells expressing higher levels of plakins tended to have longer microridges (*Figure 4—figure supplement 7*). Indeed, grouping cells into high- and low-expressing categories demonstrated that microridges in cells with high plakin levels were substantially longer than microridges in WT cells (*Figure 4D*, *Figure 4—figure supplement 7B*). Together, these results indicate that Ppl and Evpl function like a molecular rheostat for microridge length: lowering plakin expression shortens microridges whereas increasing plakin expression lengthens microridges.

## Plakins associate with microridges and keratins in zebrafish skin cells

Given their ability to bind F-actin and keratins (*Kalinin et al., 2005*; *Karashima and Watt, 2002*; *Kazerounian et al., 2002*), we hypothesized that Evpl and Ppl serve as linkers between keratins and F-actin in microridges. Previous biochemical studies showed that their N-terminal head domains can bind to actin (*Kalinin et al., 2005*), their rod domains promote dimerization (*DiColandrea et al., 2000*; *Kalinin et al., 2004*), and their C-terminal tail domains bind to keratins (*Karashima and Watt, 2002*; *Kazerounian et al., 2002*). To determine the localization of these domains in zebrafish periderm cells, we GFP- or tdTomato-tagged each domain of each plakin protein and expressed them in periderm cells of WT animals. As expected, full-length plakins localized to microridges, but the isolated head and tail domains of each plakin localized throughout the cytoplasm and nucleus, suggesting that dimerization via the rod domain is required for their localization to microridges (*Figure 5A*). The Ppl rod domain, which is required for dimerization in vitro (*Kalinin et al., 2004*), localized variably to microridges. However, the Evpl rod domain did not localize to microridges (*Figure 5A*), suggesting that it is not sufficient for dimerization, and implying that the Ppl rod domain may weakly homodimerize, as previously suggested by biochemical studies (*Kalinin et al., 2004*). Indeed, whereas full-length Ppl localized to the short microridges in *evpl* mutants (*Figure 5—figure supplement 1A*), full-length Evpl was cytoplasmic in *ppl* mutants (*Figure 5—figure supplement 1B*), consistent with the hypothesis that Ppl, but not Evpl, can homodimerize (or homo-oligomerize), and that dimerization is required for microridge localization. This observation may also explain why *ppl* mutants have a more severe microridge phenotype than *evpl* mutants (*Figure 4A–B*).

To determine where dimerized head and tail domains localize, we expressed GFP-tagged head-rod (i.e. Δtail) or rod-tail (i.e. Δhead) domain fusions. Head-rod fusions of both plakins localized to microridges more robustly than rod domains alone (*Figure 5A*), suggesting that the dimerized head domain enhances localization to microridges, potentially through F-actin binding. Ppl, but not Evpl, rod-tail domain fusions variably localized in keratin filament-like patterns (*Figure 5A*), suggesting that the dimerized Ppl tail domain associates with keratins. Since parts of the rod domain can contribute to F-actin binding (*Kalinin et al., 2005*), we made two shorter fusions containing only part of the Ppl rod domain and the entire Ppl tail domain (sRT1-GFP and sRT2-GFP). These shorter fusions were strongly localized in keratin-like patterns that included the thick microridge-associated bundles and the filamentous network throughout the cell (*Figure 5B–C*, *Figure 5—video 1*). Together these findings demonstrate that dimerized plakins have the potential to link F-actin with keratins in microridges.

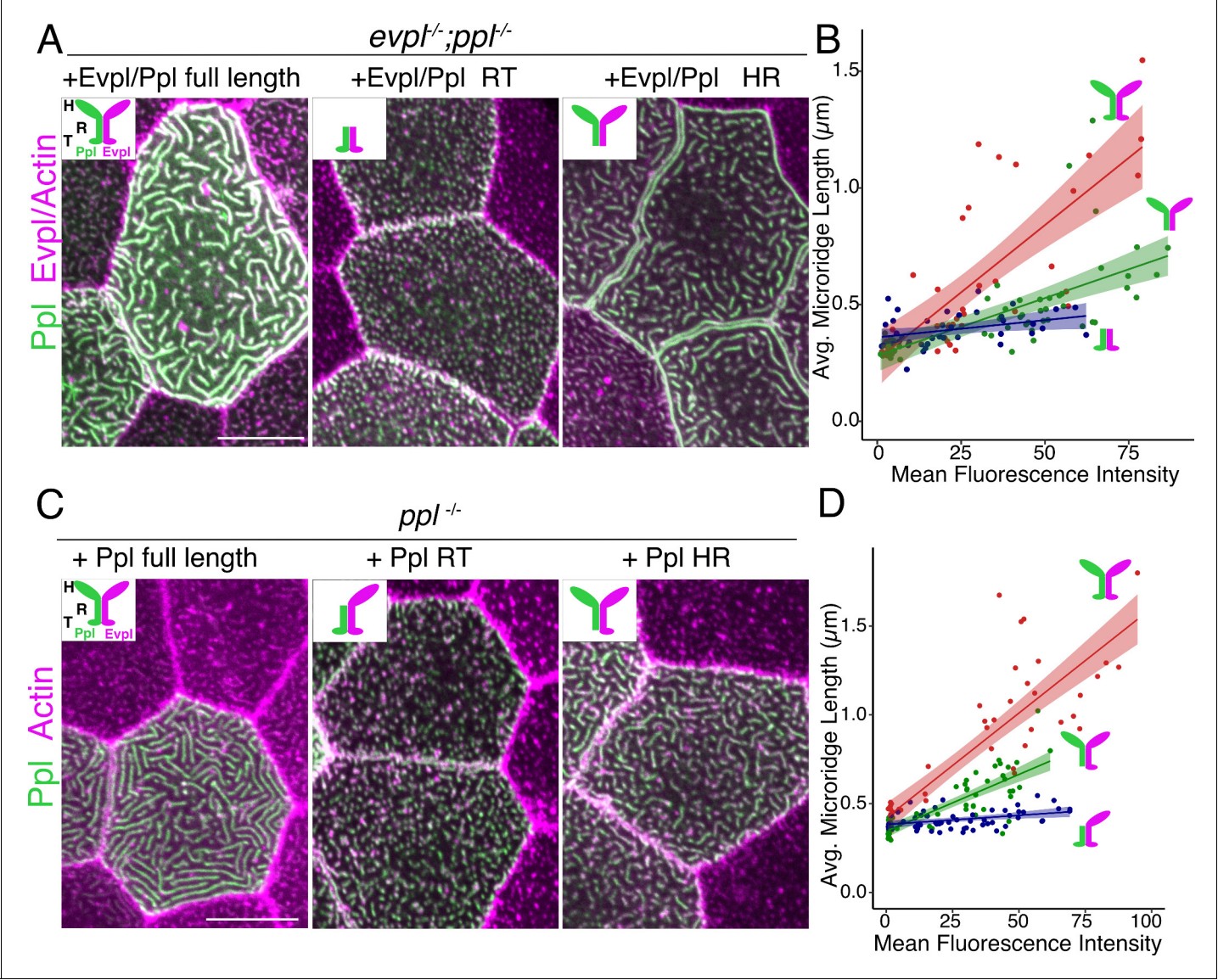

**Figure 6.** The periplakin head domain is required for initiation of microridge morphogenesis and the tail domain is required for microridge elongation. (A, C) Lifeact-mRuby-expressing cells mosaically expressing Ppl-GFP and Evpl-tdTomato variants in periderm cells, as indicated. Neighboring cells lacking GFP serve as controls. (A) Periderm cells of *evpl*<sup>−/−</sup>;*ppl*<sup>−/−</sup> mutant animals co-expressing full-length tagged Evpl and Ppl (left), tagged Evpl and Ppl rod-tail fusions (middle), or tagged Evpl and Ppl head-rod fusions (right). Inset schematics show the domains expressed. H: Head; R: Rod; T: Tail. (B) Scatter plot of average microridge length versus mean fluorescence intensity in *evpl*<sup>−/−</sup>;*ppl*<sup>−/−</sup> mutant periderm cells expressing full-length Evpl and Ppl (red line), Evpl and Ppl head-rod fusions (green line), or Evpl and Ppl rod-tail fusions (blue line). Shading indicates 95% confidence interval. Slopes = 0.0115, 0.0049, and 0.0015; R2 = 0.56, 0.44, and 0.13 for red, green, and blue lines, respectively. n = 39–55 cells from 3 to 4 fish per category. (C) Periderm cells of *ppl*<sup>−/−</sup> mutant animals co-expressing tagged Ppl full length (left), Ppl rod-tail (middle), or Ppl head-rod fusions (right). Insets illustrate the Ppl domains expressed. Green is Ppl and magenta is endogenous Evpl in diagrams. H: Head; R: Rod; T: Tail. (D) Scatter plot of average microridge length versus mean fluorescence intensity in *ppl*<sup>−/−</sup> periderm cells expressing full-length Ppl (red line), the Ppl head-rod (green line), or the Ppl rod-tail (blue line). These full-length data are also used in *Figure 7C–D*. Shading indicates 95% confidence interval. Slopes = 0.0118, 0.0064, and 0.001; R2 = 0.74, 0.67, and 0.15 of red, green, and blue lines, respectively. n = 37–57 cells from 3 to 5 fish per category. Scale bar: 10 µm.

## The periplakin head domain is required at an early step of morphogenesis to fuse pegs into microridges

To determine how each plakin domain contributes to microridge morphogenesis, we attempted to rescue plakin mutants with full-length and truncated tagged versions of the plakins. Injecting genes encoding fluorescently tagged full-length Evpl-tdTomato and Ppl-GFP into *evpl*<sup>−/−</sup>;*ppl*<sup>−/−</sup> double

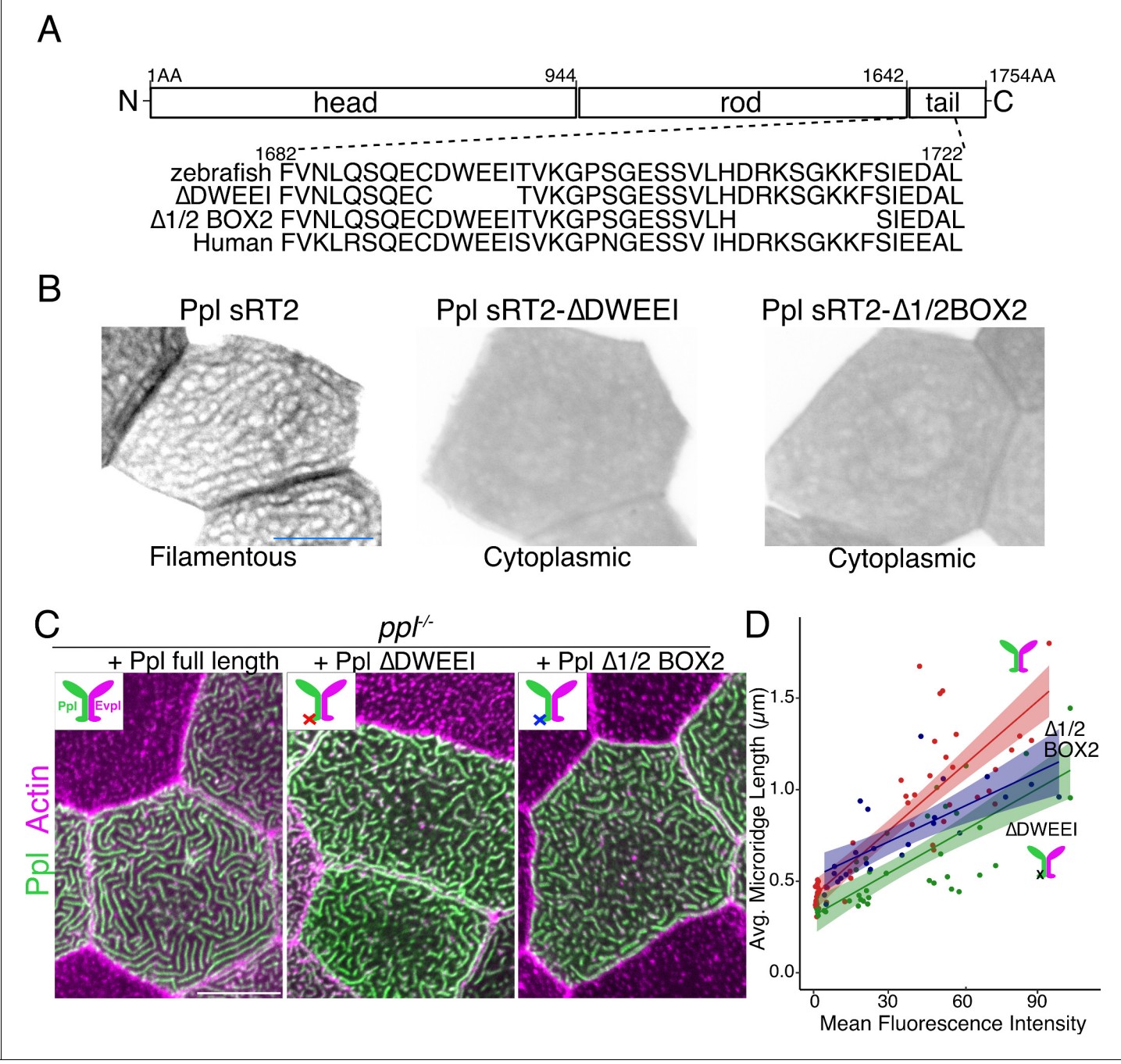

**Figure 7.** Keratin-binding domains of periplakin are required for microridge elongation. (**A**) Top: Diagram of Ppl protein domains. Amino acid numbers are indicated above. Bottom: Amino acid sequence of IF-binding domains of zebrafish and human Ppl, showing locations of the ΔDWEEI and Δ1/2BOX2 mutations. (**B**) Images of tagged Ppl(sRT2)-, Ppl(sRT2-ΔDWEEI)-, and Ppl(sRT2-Δ1/2BOX2)-expressing periderm cells in WT animals. (**C**) Lifeact-mRuby-expressing periderm cells mosaically expressing GFP-tagged Ppl (left), the Ppl(ΔDWEEI) mutant (middle), or the Ppl(Δ1/2BOX2) mutant (right) in $ppl^{-/-}$ mutant animals. Neighboring cells lacking GFP serve as controls. (**D**) Scatter plot of average microridge length versus mean fluorescence intensity in $ppl^{-/-}$ periderm cells expressing full-length Ppl (red line), Ppl(ΔDWEEI) (green line), or Ppl(Δ1/2BOX2) (blue line). Shading indicates 95% confidence interval. Slopes = 0.0118, 0.0077, and 0.0064; R = 0.74, 0.6, and 0.54, for red, green, and blue lines, respectively. n = 24–51 cells from 3 to 4 fish per category. Scale bars: 10 μm.

mutant fish rescued microridge development (*Figure 6A*). Strikingly, the average microridge length per cell correlated with fluorescence intensity (*Figure 6B*), further illustrating that plakin expression levels determine microridge length. By contrast, truncated fusion proteins lacking the plakin head domains did not rescue microridge length in double mutants (*Figure 6A–B*). Similarly, Ppl rod-tail fusions did not rescue Ppl single mutants (*Figure 6C–D*), suggesting that heterodimers containing only the Evpl head domain cannot support microridge morphogenesis.

### The keratin-binding periplakin tail domain is required to elongate microridges

To ask if plakin-keratin interactions play a role in microridge morphogenesis, we expressed tagged Evpl and Ppl head-rod domain fusions, which lack the keratin-binding tail domain, in double mutant

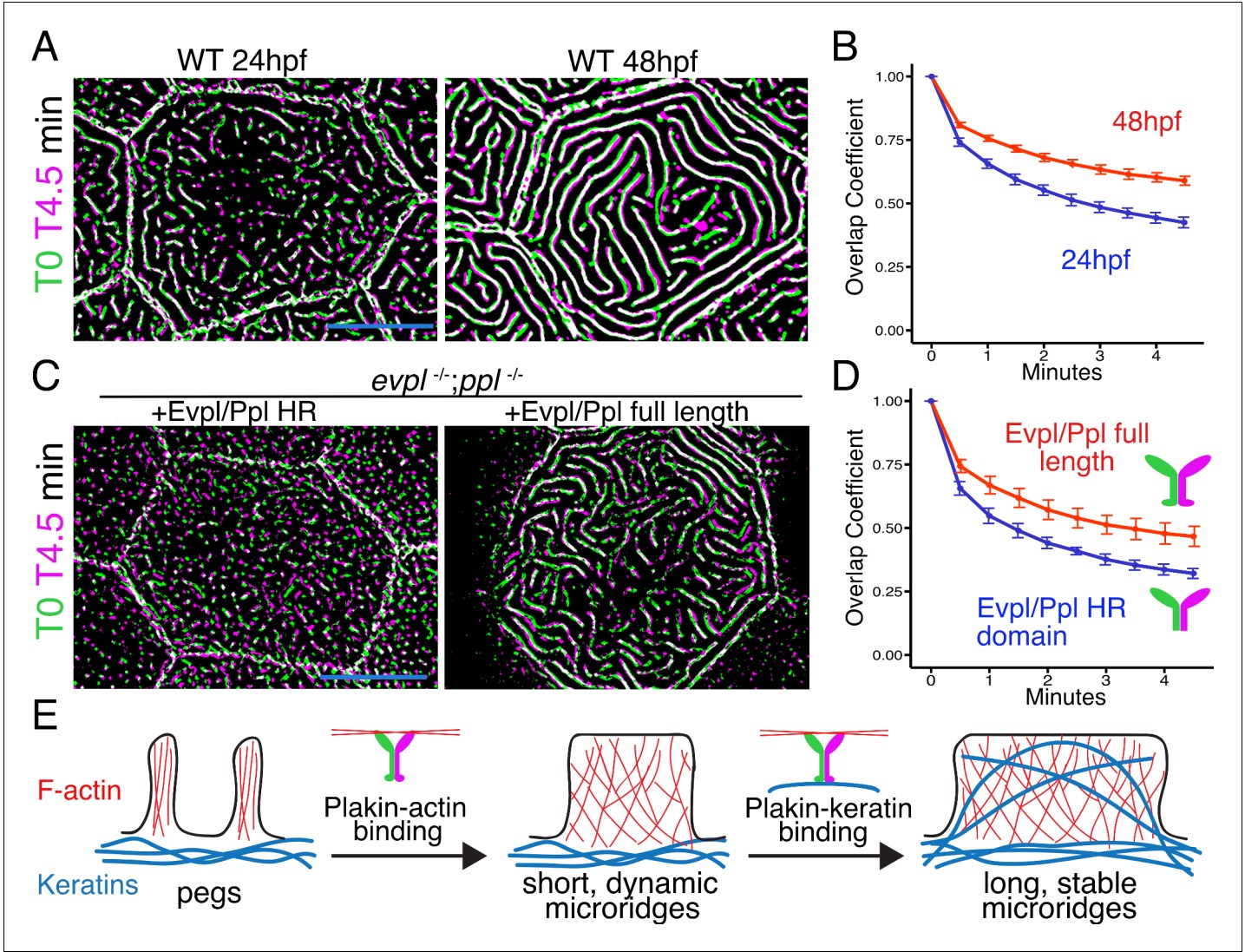

**Figure 8.** The periplakin tail domain is required to stabilize microridges. (**A**) Superimposition of first and last frames from a time-lapse movie of WT cells expressing Lifeact-GFP at the indicated developmental stages. Green: Starting time point, Magenta: 4.5 min time point. Overlap is white. (**B**) Line plots of overlap coefficients at the indicated stages, comparing each time point to the first. n = 21–33 cells from 4 to 5 fish per category. Bars = 95% confidence interval. (**C**) Superimposition of first and last frames from a time-lapse movie of *evpl⁻/⁻;ppl⁻/⁻* double mutant animals expressing Lifeact-GFP, rescued with the indicated Evpl and Ppl fusions. (**D**) Line plots of overlap coefficients in cells expressing the indicated Evpl and Ppl fusions, comparing each time point to the first. n = 11 cells from 3 to 4 fish per category. Bars = 95% confidence interval. (**E**) Two-step model of microridge morphogenesis. First, plakins interact with F-actin via their head domains to fuse pegs into short microridges. Second, plakin-keratin interactions through plakin tail domains stabilize microridges, allowing them to further elongate. Scale bars: 10 μm.

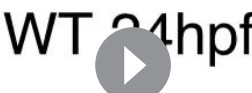

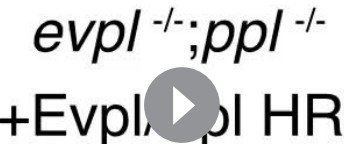

**Video 3.** Microridges are more stable at later developmental stages. Time-lapse confocal videos of periderm cells expressing Lifeact-mRuby at 24 and 48hpf, as indicated by title cards. Time stamps in upper left.
https://elifesciences.org/articles/58149#video3

**Video 4.** The periplakin tail domain is required to stabilize microridges. Time-lapse confocal videos of $evpl^{-/-};ppl^{-/-}$ periderm cells expressing Lifeact-RFP and either (1) GFP-tagged Ppl head-rod (i.e. Δtail) and tdTomato-tagged Evpl head-rod, or (2) GFP-tagged full-length Ppl and tdTomato-tagged full-length Evpl, as indicated by title cards. Time stamps in upper left.
https://elifesciences.org/articles/58149#video4

embryos. These fusions rescued the initiation of microridge formation but did not rescue microridge length as well as full-length plakins (*Figure 6A–B*). These results indicate that the Evpl and Ppl dimerized head domains are sufficient for the initiation of microridge morphogenesis but not for their full elongation. Similarly, expressing the Ppl head-rod fusion in *ppl* single mutants allowed the formation of only short, but not fully elongated, microridges (*Figure 6C–D*). Together these experiments suggest that plakin-keratin association facilitates microridge elongation.

To further test if plakin-keratin interactions promote microridge elongation, we deleted five amino acids in the Ppl tail domain required for keratin binding (ΔDWEEI) (*Figure 7A*; *Karashima and Watt, 2002*). Localization of Ppl(ΔDWEEI) rod-tail fusions (sRT2[ΔDWEEI]-GFP) to the keratin network was severely reduced (*Figure 7B*), confirming in vivo that this site is required for optimal Ppl-keratin binding. Similar to head-rod fusions, Ppl(ΔDWEEI) could not rescue microridge length in *ppl* mutants as well as WT Ppl. Deleting another small domain required for keratin binding (Δ1/2Box2) (*Karashima and Watt, 2002*) in Ppl yielded similar results (*Figure 7C–D*). These findings suggest that Ppl-keratin interactions are required for full microridge elongation.

## Plakin-keratin interactions stabilize microridges

IFs are the strongest cytoskeletal elements (*Janmey et al., 1991*), maintain their structure even as they replace subunits (*Colakoğlu and Brown, 2009*), and protect cells from mechanical stress (*Leube et al., 2017*). We, therefore, hypothesized that plakin-mediated recruitment of keratins into microridges stabilizes them, and that stabilization permits their elongation. To test our hypothesis, we first compared the stability of short, nascent microridges at early development (24hpf), when keratins are not found abundantly in microridges, to their stability at a later stage (48hpf), when microridges are longer and contain keratins. Over the course of 5 minute time-lapse movies, the microridge pattern changed more rapidly at 24hpf than at 48hpf, indicating that shorter microridges likely lacking keratins are less stable than longer keratin-containing microridges in older animals (*Figure 8A–B*, *Video 3*). To test if plakin-keratin binding plays a role in stabilization, we compared cells in double mutant animals expressing full-length Evpl and Ppl to those expressing the Evpl and Ppl head-rod domains (i.e. Δtail domain) at 48hpf. Microridges were more dynamic in cells

expressing Evpl and Ppl head-rod fusions than those expressing full-length plakins (*Figure 8C–D*, *Video 4*), similar to microridges in WT cells of younger animals. These results suggest that plakin-keratin interactions stabilize microridges to allow them to elongate.

## Discussion

This study has uncovered new roles for cytoskeletal filaments and cytolinkers in cellular morphogenesis, dissecting a complex morphogenetic process into discrete, mechanistically distinct steps. Of the three classes of cytoskeletal elements, only IFs were not previously thought to directly contribute to the morphogenesis of cellular protrusions. However, our findings suggest that keratin IFs play a key role in stabilizing and elongating microridge epithelial protrusions. Plakin proteins, which can bind to both actin and keratin filaments, are required and sufficient for microridge elongation, implying that this process involves the integration of two distinct cytoskeletal components. In contrast to other membrane protrusions, which grow and shrink as a single unit, microridges form from the coalescence of peg precursors (*Lam et al., 2015*; *van Loon et al., 2020*). Dissecting plakin functions in microridge development uncovered an additional, cryptic step of morphogenesis—the elongation of short, dynamic microridges into long, stable microridges. Thus, microridge morphogenesis proceeds through at least three molecularly separable steps: (1) peg formation, (2) peg coalescence to form nascent microridges, and (3) microridge elongation (*Figure 8E*).

Evpl and Ppl are first required for the coalescence of pegs into microridges, since periderm cells in *evpl* and *ppl* mutants retain pegs, but microridges are reduced or absent in these mutants (*Figure 4A–B*). A role for the plakins at this step of morphogenesis was also supported by the observation that Ppl formed longer and more continuous structures than F-actin during the process of peg coalescence (*Figure 3F–G*). This observation suggests the intriguing possibility that the plakins, which can form oligomers on their own in vitro (*Kalinin et al., 2004*), guide peg coalescence. Plakins lacking their N-terminal head domains failed to rescue peg coalescence in plakin mutants (*Figure 6A–B*), indicating that this protein region is required at this early morphogenetic step. The plakin head domains have direct actin-binding activity (*Jefferson et al., 2004*; *Kalinin et al., 2005*; *Sonnenberg and Liem, 2007*) and also contain spectrin repeats and a SRC homology domain 3 (SH3) domain (*Jefferson et al., 2004*; *Sonnenberg and Liem, 2007*), which in other proteins can bind to actin interacting proteins, including the Arp2/3 activator WAVE (*Cestra et al., 2005*). Peg coalescence requires not only the plakin proteins, but also the Arp2/3 actin branch nucleating complex (*Lam et al., 2015*; *Pinto et al., 2019*; *van Loon et al., 2020*) and actomyosin contraction (*van Loon et al., 2020*). Thus, it is likely that Evpl and Ppl contribute to peg coalescence by interacting directly with F-actin or actin regulators, helping to incorporate the actin bundles within pegs into the larger actin network of microridges.

The next step of microridge morphogenesis, the elongation of dynamic nascent microridges into mature microridges, was revealed by removing the C-terminal domains of Evpl and Ppl (*Figure 6A–B*), and by mutating a few amino acids in Ppl required for IF binding (*Figure 7*). Without this keratin-binding domain, plakins could promote peg coalescence, but not full microridge elongation, indicating that Evpl and Ppl's role in the earlier step is independent of keratin-binding. Consistent with this notion, we detected keratin in microridges only at later stages, when microridges were longer (*Figure 1C–E*). We thus propose that Evpl and Ppl recruit keratins into microridges to instigate the subsequent elongation step of morphogenesis (*Figure 8E*). Evpl and Ppl were both necessary and sufficient for lengthening microridges (*Figure 4*), and, remarkably, microridge length correlated tightly with plakin expression levels (*Figure 6*). The fact that overexpressing Krt17 also lengthened microridges (*Figure 1F*) lends support to the notion that plakins determine microridge length by recruiting keratins.

How do keratins promote microridge elongation? Our observation that short protrusions, which tend to lack keratin, are less stable than long microridges, which contain keratin (*Figure 8A–B*), suggests that keratin stabilization of microridges might permit their elongation. Indeed, although plakin proteins lacking IF-binding domains enabled the formation of short, relatively unstable microridges, they never matured into long, stable structures (*Figure 8C–D*). Perhaps most compellingly, overexpressing Krt17 preserved the three-dimensional structure of microridges upon F-actin disruption (*Figure 2A–C*). Together these observations indicate that keratins provide microridges with stability. Notably, the IF vimentin has also been hypothesized to play a role in the growth of invadopodia at a

late morphogenetic step (*Schoumacher et al., 2010*). Perhaps vimentin, similar to keratin in microridges, stabilizes invadopodia to allow them to elongate.

A role for IFs as microridge stabilizers is consistent with the fact that IFs are intrinsically stronger than the other cytoskeletal elements, and can preserve their form even as subunits are replaced (*Colakoğlu and Brown, 2009*; *Janmey et al., 1991*). By contrast, actin filaments are more dynamic, consistent with their role in forming transient protrusions, like lamellipodia, filopodia, and dorsal ruffles. Microridges remodel but do not form and disassemble as rapidly as the aforementioned protrusions. In terms of stability, microridges may be more similar to stereocilia or microvilli, although each of these structures likely use different strategies to maintain their forms. The extreme stability of stereocilia reflects the fact that there is little F-actin turnover within them (*Narayanan et al., 2015*; *Zhang et al., 2012*), whereas microvilli maintain their morphologies despite constant actin turnover (*Loomis et al., 2003*; *Meenderink et al., 2019*; *Tyska and Mooseker, 2002*). Like microvilli, microridges constantly replace their F-actin scaffolds, since inhibiting Arp2/3 leads to disassembly of the microridge F-actin network within thirty minutes (*Lam et al., 2015*; *van Loon et al., 2020*). However, in contrast to microvilli, microridges require keratins to preserve their forms in the face of F-actin turnover, perhaps because their more extended morphologies and less organized F-actin networks (*Pinto et al., 2019*) require additional stabilization. A landmark study comparing the three classes of cytoskeletal elements speculated that the mechanical 'differences between F-actin and vimentin are optimal for the formation of a composite material with a range of properties that cannot be achieved by a single polymer network' (*Janmey et al., 1991*). By combining F-actin with IFs, microridges may achieve an optimal balance between plasticity and stability.

## Materials and methods

### Zebrafish

Zebrafish (*Danio rerio*) were grown at 28.5℃ on a 14 hr/10 hr light/dark cycle. Embryos were raised at 28.5℃ in embryo water (0.3 g/L Instant Ocean salt, 0.1% methylene blue). For live confocal imaging, pigmentation was blocked by treating embryos with phenylthiourea (PTU) at 24hpf. All experimental procedures were approved by the Chancellor's Animal Research Care Committee at UCLA.

### CRISPR/Cas9 mutagenesis

To generate guide RNAs (gRNAs) we used the 'short oligo method to generate gRNA', as previously described (*Talbot and Amacher, 2014*). Two Cas9 binding sites were selected for each gene. *Evpl* targeting sequences were located in exons 3 and 7, *ppl* targeting sequences were in exons 2 and 5. The DNA template was PCR amplified to make a product containing a T7 RNA polymerase promoter, the gene targeting sequence, and a gRNA scaffold sequence. PCR products were used as a template for RNA synthesis with T7 RNA polymerase (New England Biolabs) and purified (QIAGEN RNA purification kit) to generate gRNAs. Injection mixes contained Cas9 protein (1 mg/mL; IDT), gRNAs (0.5–1 ng/µL), and 300 mM KCl. Injection mixes were incubated on ice 15 min before injection. Embryos were injected at the 1-cell stage with 2–5 nL of injection mix. To identify germline founders, F0 fish were crossed to wild-type fish and 48hpf embryos were collected for PCR genotyping. Founder progeny were raised to adulthood to establish stable mutant lines.

### Reverse transcription PCR (RT-PCR)

RNA was extracted with TRIzol (Thermo FIsher Scientific) from five scales per adult for each genotype. RNA was purified (QIAGEN RNA purification kit), reverse transcribed with Superscript three using an Oligo(dT)20 primer to make cDNA (Invitrogen), and PCR amplified. Primers for RT-PCR are listed in the Key Resources Table.

### Morpholino knockdown and rescue

About 4 ng of morpholino antisense oligonucleotides (Gene Tools), targeting splice sites in *evpl* or *ppl*, were injected into 1-cell stage embryos. A GFP-tagged cDNA for *evpl* and a tdTomato-tagged cDNA for *ppl* were injected at the 1-cell stage for rescue experiments. *evpl* MO sequence: 5'-GTG TCTTTAGTGTCACTCACTCATT-3'; *ppl* MO sequence: 5'-TCTGAGGTGAAACAACAGCGAGTTT-3' (also listed in the Key Resources Table).

## BAC reporters

Tg(Krt5:Lifeact-GFP)[LA226] and Tg(Krt5:Lifeact-mRuby)[LA227] lines were previously described (*van Loon et al., 2020*). To create translational fusion transgenes, GFP or mRuby reporter gene cassettes were recombined into a site directly preceding the stop codon of target genes in bacterial artificial chromosomes (BACs), as previously described (*Cokus et al., 2019*; *Suster et al., 2011*). BAC identifiers are listed in the Key Resources Table.

## Plasmids

Plasmids were constructed using the Gateway-based Tol2kit (*Kwan et al., 2007*). Primer sequences are listed in the Key Resources Table. The following plasmids were previously described: p5E-Krt5 (*Rasmussen et al., 2015*), pME-EGFPpA, p3E-EGFPpA, p3E-tdTomato, and pDestTol2pA2 (*Kwan et al., 2007*), Krt5-Lifeact-GFP and Krt5-Lifeact-mRuby (*van Loon et al., 2020*).

Krt5:Ppl(ΔDWEEI) and Krt5:Ppl(Δ1/2Box2) transgenes were created with PCR from Krt5-Ppl-GFP or Krt5-Ppl(sRT2)-GFP plasmids with SuperFi DNA Polymerase (Invitrogen). PCR products were gel extracted and transformed, and selected colonies were sequenced.

To create endogenously tagged Ppl and Krt17 alleles in transient transgenics, CRISPR gRNA target sites were selected 676 bp (Ppl) and 895 bp (Krt17) downstream of the stop codon. Donor plasmids for recombination were generated using the Gateway-based Tol2kit. These plasmids contained a 5' homology arm consisting of an ~1 Kb sequence upstream of the stop codon, GFP with a polyA termination sequence from EGFP-SV40, and a 3' homology arm consisting of an 800 bp (Krt17) or 1 Kb (Ppl) sequence downstream of the gRNA target site. Primer sequences used to amplify homology arms are listed in the Key Resources Table. To linearize the donor plasmid, a gRNA with a target site on the plasmid was created (see primers in the Key Resources Table); 2–5 nL of this mix were injected into 1-cell stage embryos and fluorescence was observed with confocal microscopy at 24hpf. Injection mixes contained Cas9 mRNA (250 ng/μL), gRNAs for each gene (25 ng/μL), a gRNA for the donor plasmid (25 ng/μL), and the donor plasmid (25 ng/μL). Cas9 mRNA was synthesized as previously described (*Julien et al., 2018*).

## Mounting embryos for live imaging

Live zebrafish embryos were anaesthetized with ~0.2 mg/mL MS-222 (tricaine) in system water prior to mounting. Embryos were embedded in 1.2% agarose on a cover slip (Fisher Scientific) and a plastic ring was sealed with vacuum grease onto the coverslip to create a chamber that was filled with 0.2 mg/mL MS-222 solution, as previously described (*O'Brien et al., 2009*). High precision cover glasses (Marienfeld) were used for Airyscan and Elyra microscopy.

## Microscopy

Confocal imaging was performed on an LSM 800 or LSM 880 microscope with Airyscan (Carl Zeiss) using a 40× oil objective (NA = 1.3) or 60× oil objective (NA = 1.4). SIM imaging was performed on an Elyra microscope (Carl Zeiss) using a 60× oil objective (NA = 1.4).

## Drug treatment

CK666 (Fisher Scientific) was dissolved in DMSO. Treatment solutions were created by adding CK666 or an equivalent volume of DMSO (≤2%) to Ringer's Solution with 0.2 mg/mL MS-222. Zebrafish larvae were treated with 200 μM CK666 or DMSO just before imaging. During imaging, larvae were mounted in agarose in sealed chambers, as described above, and chambers were filled with treatment solutions.

## Phalloidin staining of adult scales

Fish were anesthetized in 0.016% MS-222 (wt/vol) dissolved in system water to remove scales. Scales were removed from the lateral trunk region of 3 month old fish with forceps. Isolated scales were fixed in 4% PFA for 30 min at room temperature on a shaker. Scales were washed twice in 0.01% Tween in PBS (PBST), then permeabilized for 10 min at room temperature with 0.1% TritonX-100 in PBS. Scales were incubated for 2 hr at room temperature with AlexaFluor 488 phalloidin (Thermo Fisher Scientific) diluted 1:250 in PBST. Scales were washed 2 × 10 min with PBST on a shaker,

mounted inside reinforcement labels (Avery 5722) on a slide, and filled with PBST. The coverslips (Fisher Scientific) were sealed with nail polish over the reinforcement labels.

## Image analysis and statistics

Image analysis was performed with FIJI (*Schindelin et al., 2012*). For display purposes, confocal z-stack images were projected (maximum intensity projection) and brightness and contrast were optimized. The Image Stabilizer plugin was used to adjust for cell drift. An automated pipeline implemented in FIJI was used to analyze average microridge length per cell, as previously described (*van Loon et al., 2020*).

To analyze fluorescence intensity, all images were acquired with identical imaging parameters. Cells were outlined by hand, and the background was subtracted using the 'rolling ball' radius 50.0 pixels method. The area outside cells was cleared before the mean fluorescence intensity was measured.

To analyze overlap coefficients, cells were outlined by hand, brightness and contrast were automatically enhanced, and the area around cells was cleared. Lifeact-GFP images were blurred using the Smoothen function three times, and passed through a Laplacian morphological filter from the MorphoLibJ plugin, using the square element and a radius of 1, as previously described (*van Loon et al., 2020*). Images were thresholded using the Triangle method for Lifeact-GFP images and the Percentile method for Krt17-GFP images. Thresholded images were analyzed to obtain overlap coefficients using the JACoP FIJI plugin.

Statistical analyses and graphs were generated with RStudio. Details of statistics for each experiment are listed in Figure Legends.

## Acknowledgements

We thank Sally Horne-Badovinac, Manish Butte, Jeff Rasmussen, Margot Quinlan, and members of the Sagasti lab for comments on the manuscript, Khaled Nassman for help with analysis, Son Giang and Linda Dong for excellent fish care, and Nat Prunet of the MCDB/BSCRC microscopy core for help with microscopy. APvL was supported by a Ruth L Kirschstein National Research Service Award (GM007185); this study was funded by NIH grants R21EY024400 and R01GM122901 to AS.

## Additional information

### Funding

| Funder | Grant reference number | Author |
|--------|------------------------|--------|
| NIGMS | R01GM122901 | Alvaro Sagasti |
| NEI | R21EY024400 | Alvaro Sagasti |
| NIGMS | GM007185 | Aaron Paul van Loon |

The funders had no role in study design, data collection and interpretation, or the decision to submit the work for publication.

### Author contributions

Yasuko Inaba, Conceptualization, Resources, Formal analysis, Supervision, Validation, Investigation, Visualization, Methodology, Writing - original draft, Writing - review and editing; Vasudha Chauhan, Investigation, Methodology; Aaron Paul van Loon, Formal analysis, Visualization, Methodology; Lamia Saiyara Choudhury, Investigation; Alvaro Sagasti, Conceptualization, Supervision, Funding acquisition, Visualization, Writing - original draft, Writing - review and editing

### Author ORCIDs

Yasuko Inaba (iD) https://orcid.org/0000-0002-3239-2075
Alvaro Sagasti (iD) https://orcid.org/0000-0002-6823-0692

### Ethics

Animal experimentation: All animal experimental procedures were approved by the Chancellor's Animal Research Care Committee at UCLA (protocol #2005-117-41D).

### Decision letter and Author response

Decision letter https://doi.org/10.7554/eLife.58149.sa1
Author response https://doi.org/10.7554/eLife.58149.sa2

## Additional files

### Supplementary files

• Supplementary file 1. Tables of p-values for microridge length and area comparisons for *Figure 4*. (A) Table of p-values comparing microridge length per cell between the indicated genotypes (as in *Figure 4B*) with the Wilcoxon rank-sum test. (B) Table of p-values comparing microridge length per cell, averaged for each animal (see *Figure 4—figure supplement 2A*), between the indicated genotypes with the Wilcoxon rank-sum test. (C) Table of p-values comparing pooled microridge length distributions between the indicated genotypes (as in *Figure 4—figure supplement 2B*) with the Wilcoxon rank-sum test. (D) Table of p-values comparing microridge cell areas between the indicated genotypes (as in *Figure 4—figure supplement 2C*) with the Wilcoxon rank-sum test. Black font indicates p-values>0.05 and red font indicates p-values<0.05.

• Transparent reporting form

### Data availability

All data generated or analyzed during this study are included in the manuscript and supporting files.

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

# Appendix 1

**Appendix 1—key resources table**

| Reagent type (species) or resource | Designation | Source or reference | Identifiers | Additional information |
|---|---|---|---|---|
| Genetic reagent (zebrafish; *Danio rerio*) | Tg(krt5:Lifeact-GFP) | *van Loon et al., 2020* | la226Tg | |
| Genetic reagent (zebrafish; *Danio rerio*) | Tg(Krt5:Lifeact-Ruby) | *van Loon et al., 2020* | la227Tg | |
| Genetic reagent (zebrafish; *Danio rerio*) | Tg(UAS:eGFP-PH-PLC), a.k.a Tg(4xuAS:UAS:eGFP-Rno-Plcd1) | *Jiang et al., 2019* | la216Tg | |
| Genetic reagent (zebrafish; *Danio rerio*) | *Envoplakin* mutant | See *Figure 4*, this paper | | |
| Genetic reagent (zebrafish; *Danio rerio*) | *Periplakin* mutant | See *Figure 4*, this paper | | |
| Recombinant DNA reagent | Bacterial artificial chromosome | CHORI-211 BAC library | Ch211-188J19 | BAC for creating Envoplakin reporter |
| Recombinant DNA reagent | Bacterial artificial chromosome | CHORI-73 BAC library | Ch73-360K2 | BAC for creating Periplakin reporter |
| Recombinant DNA reagent | Bacterial artificial chromosome | CHORI-73 BAC library | Ch73-107M18 | BAC for creating Krt17 reporter |
| Recombinant DNA reagent | Bacterial artificial chromosome | CHORI-73 BAC library | Ch73-294H4 | BAC for creating Krt99 reporter |
| Recombinant DNA reagent | Bacterial artificial chromosome | CHORI-73 BAC library | Ch73-107M18 | BAC for creating Krt92 reporter |
| Recombinant DNA reagent | Bacterial artificial chromosome | CHORI-73 BAC library | Ch73-107M18 | BAC for creating krt1-19d reporter |
| Recombinant DNA reagent | Bacterial artificial chromosome | CHORI-73 BAC library | Ch73-107M18 | BAC for creating cyt1 reporter |
| Recombinant DNA reagent | Bacterial artificial chromosome | CHORI-73 BAC library | Ch73-107M18 | BAC for creating cyt1l reporter |
| Sequence-based reagent | Morpholino antisense oligonucleotide targeting Ppl | Gene Tools, LLC | | 5'-TCTGAGG TGAAACAACAGCGAGTTT-3 |
| Sequence-based reagent | Morpholino antisense oligonucleotide targeting Evpl | Gene Tools, LLC | | 5'-GTGTCTTTAGTGTCACTCAC TCATT-3' |
| Sequence-based reagent | Genotyping Evpl wild-type and mutant allele; forward primer | Fisher Scientific | | 5'-AGAAGAGCTGCTGCTCC TTG-3' |
| Sequence-based reagent | Genotyping Evpl wild-type allele; reverse primer | Fisher Scientific | | 5'-CTCATGAAGGCGAC TTACAC-3' |
| Sequence-based reagent | Genotyping Evpl mutant allele; reverse primer | Fisher Scientific | | 5'-TATTGTCTGCGAACG TCAGC-3' |

*Appendix 1—key resources table continued*

| Reagent type (species) or resource | Designation | Source or reference | Identifiers | Additional information |
|---|---|---|---|---|
| Sequence-based reagent | Genotyping Ppl wild-type and mutant allele; forward primer | Fisher Scientific | | 5'-TTCCCTCAGCAGATGA TGCG-3' |
| Sequence-based reagent | Genotyping Ppl wild-type allele; reverse primer | Fisher Scientific | | 5'-C TCAGCAAGAGAGCCGACAC-3' |
| Sequence-based reagent | Genotyping Ppl mutant allele; reverse primer | Fisher Scientific | | 5'-CCAGTAGTCTGTTGTACC TCGC-3' |
| Sequence-based reagent | Forward primer for evpl exon3 gRNA synthesis | Fisher Scientific | | 5'-GCGTAATACGACTCACTA TAGG CAGCAGATCTCCTTGGCCG TTTT AGAGCTAGAAATAGC-3' |
| Sequence-based reagent | Forward primer for evpl exon7 gRNA synthesis | Fisher Scientific | | 5'-GCGTAATACGACTCACTA TAGGA GGAGGTGGCATACATGAG TTTT AGAGCTAGAAATAGC-3' |
| Sequence-based reagent | Forward primer for ppl exon2 gRNA synthesis | Fisher Scientific | | 5'-GCGTAATACGACTCACTA TAGGCG CCAAATCAGTTTCGGGGTTTT AGAGCTAGAAATAGC-3' |
| Sequence-based reagent | Forward primer for ppl exon5 gRNA synthesis | Fisher Scientific | | 5'-GCGTAATACGACTCACTA TAGGATA GTGAAGTGGAGGCACGTTTT AGAGCTAGAAATAGC-3' |
| Sequence-based reagent | Forward primer for Ppl-GFP homologous recombination gRNA synthesis | Fisher Scientific | | 5'-CGCTAATACGACTCACTA TAGGGGT CTATTATATGCGGTCTAG TTTTAGAGC TAGAAATAGC-3' |
| Sequence-based reagent | Forward primer for Krt17-GFP homologous recombination gRNA synthesis | Fisher Scientific | | 5'-CGCTAATACGACTCACTA TAGGGTGG CGATTCATTCCGTTGGTTTT AGAGCTAGAAATAGC-3' |
| Sequence-based reagent | Forward primer of donor plasmid gRNA synthesis for homologous recombination | Fisher Scientific | | 5'-CGCTAATACGACTCACTA TAGGATAGTG AAGTGGGTAGA TGGACCGGGGGTTTTAGA GCTAGAAATAGC-3' |
| Sequence-based reagent | Constant reverse primer for gRNA synthesis | Fisher Scientific | | 5'-AAAGCACCGACTCGG TGCCACTTTTT CAAGTTGATAACGGACTAGCC TTATTTTAA CTTGCTATTTCTAGCTC TAAAAC-3' |
| Sequence-based reagent | Forward primer for Ppl homology arm upstream of the stop codon | Fisher Scientific | | 5'-GGGGACAACTTTGTA TAGAAA GTTGGCGCGGCTTAATGAAC TTG-3' |
| Sequence-based reagent | Reverse primer for Ppl homology arm upstream of the stop codon | Fisher Scientific | | 5'-GGGGACTGCTTTTTTG TACAAA CTTGCTTTGCCTCTGCCTGA-CA-3' |

*Continued on next page*

*Appendix 1—key resources table continued*

| Reagent type (species) or resource | Designation | Source or reference | Identifiers | Additional information |
|---|---|---|---|---|
| Sequence-based reagent | Forward primer for Ppl homology arm downstream of the stop codon | Fisher Scientific | | 5'-GGGGACAGCTTTCTTG TACAAA GTGGGATCACCACTTGACCTG TG-3' |
| Sequence-based reagent | Reverse primer for Ppl homology arm downstream of the stop codon | Fisher Scientific | | 5'-GGGGACAACTTTGTATAA TAAA GTTGCTTAATTGTTTATGTGG TATCC-3' |
| Sequence-based reagent | Forward primer for Krt17 homology arm upstream of the stop codon | Fisher Scientific | | 5'-GGGGACAACTTTGTA TAGAAA GTTGGCATTCATTAATAGTTCA- CATTC-3' |
| Sequence-based reagent | Reverse primer for Krt17 homology arm upstream of the stop codon | Fisher Scientific | | 5'-GGGGACTGCTTTTTTGTA CAAACTTGCAGTGGTTTTAGTC TGGC-3' |
| Sequence-based reagent | Forward primer for Krt17 homology arm downstream of the stop codon | Fisher Scientific | | 5'-GGGGACAGCTTTCTTG TACAAA GTGGGAACTATCTGTAGCATC TCG-3' |
| Sequence-based reagent | Reverse primer for Krt17 homology arm downstream of the stop codon | Fisher Scientific | | 5'-GGGGACAACTTTGTATAA TAAA GTTGCCTTTGGACTGTAAGG- GA-3' |
| Sequence-based reagent | Forward primer of Evpl head domain | Fisher Scientific | | 5'-GGGGACAAGTTTG TACAAAAAAGCA GGCTTTATGTTCAAGAGGAA TAAAGACA-3' |
| Sequence-based reagent | Reverse primer of Evpl head domain | Fisher Scientific | | 5'-GGGGACCACTTTG TACAAGAAAGCT GGGTAGAGAACTTCAGTGTCC TCAAAG-3' |
| Sequence-based reagent | Forward primer of Evpl rod domain | Fisher Scientific | | 5'-GGGGACAAGTTTG TACAAAAAA GCAGGCTTTATGCAACAG- CAGCTGC-3' |
| Sequence-based reagent | Reverse primer of Evpl rod domain | Fisher Scientific | | 5'-GGGGACCACTTTG TACAAGAAA GCTGGGTATGTGGAAACATT GACTTCTTTTG-3' |
| Sequence-based reagent | Forward primer of Evpl tail domain | Fisher Scientific | | 5'-GGGGACAAGTTTG TACAAAAAA GCAGGCTTTATGGACTCCTC TGAGATGCG-3' |
| Sequence-based reagent | Reverse primer of Evpl tail domain | Fisher Scientific | | 5'-GGGGACCACTTTG TACAAGAAA GCTGGGTAGACA TTTTTTGAGG AAGCATTAAACATCG-3' |
| Sequence-based reagent | Forward primer of Ppl head domain | Fisher Scientific | | 5'-GGGGACAAGTTTG TACAAAAAA GCAGGCTTTATGTTCAAGAAAA GGCAAACCA-3' |
| Sequence-based reagent | Reverse primer of Ppl head domain | Fisher Scientific | | 5'-GGGGACCACTTTG TACAAGAAA GCTGGGTACTCATCCAGTT GAGGGTCCGGC-3' |

*Continued on next page*

*Appendix 1—key resources table continued*

| Reagent type (species) or resource | Designation | Source or reference | Identifiers | Additional information |
|---|---|---|---|---|
| Sequence-based reagent | Forward primer of Ppl rod domain | Fisher Scientific | | 5'-GGGGACAAGTTTG TACAAAAAA GCAGGCTTTATGAAGGTG CCGGACCCTCAAC-3' |
| Sequence-based reagent | Reverse primer of Ppl rod domain | Fisher Scientific | | 5'-GGGGACCACTTTG TACAAGAAA GCTGGGTACACTCGCATTGC AGCCAGACTC-3' |
| Sequence-based reagent | Forward primer of Ppl tail domain | Fisher Scientific | | 5'-GGGGACAAGTTTG TACAAAAAA GCAGGCTTTATGAAGAGTCT GGCTGCAATGC-3' |
| Sequence-based reagent | Reverse primer of Ppl tail domain | Fisher Scientific | | 5'-GGGGACCACTTTG TACAAGAAA GCTGGGTATTTGCCTCTGCC TGACAGCATG-3' |
| Sequence-based reagent | Forward primer of Ppl short rod-tail1 domain | Fisher Scientific | | 5'-GGGGACAAGTTTG TACAAAAAA GCAGGCTGGATGGATGCGGA GCTCGAAATTCAG-3' |
| Sequence-based reagent | Forward primer of Ppl short rod-tail2 domain | Fisher Scientific | | 5'-GGGGACAAGTTTG TACAAAAAA GCAGGCTGGATGATCACAA- CAAC AGAAACCAGACAC-3' |
| Sequence-based reagent | Constant reverse primer of Ppl short rod-tail domain | Fisher Scientific | | 5'-GGGGACCACTTTG TACAAGAAA GCTGGGTAGACA TTTTTTGAGG AAGCATTAAACATCG-3' |
| Sequence-based reagent | Forward primer of Ppl ΔDWEEI | Fisher Scientific | | 5'-ACCGTCAAGGGTCCAAG TGGC-3' |
| Sequence-based reagent | Reverse primer of Ppl ΔDWEEI | Fisher Scientific | | 5'-GCACTCTTGACTTTGAA GGTTGACAAAC-3' |
| Sequence-based reagent | Forward primer of Ppl RT-PCR | Fisher Scientific | | 5'-AACCAAGAG TAGCGCGACCATC-3' |
| Sequence-based reagent | Reverse primer of Ppl RT-PCR | Fisher Scientific | | 5'-AGTAGTCTGTTGTACC TCGCC −3' |
| Sequence-based reagent | Forward primer of Epl RT-PCR | Fisher Scientific | | 5'-TGAGAAAGATGTAC TGCGCG −3' |
| Sequence-based reagent | Forward primer of Epl RT-PCR | Fisher Scientific | | 5'-CGTCAGCATGGTCGACCA TC-3' |
| Sequence-based reagent | Forward primer of βactin RT-PCR | Fisher Scientific | | 5'-CGTGACATCAAGGAGAAGC T-3' |
| Sequence-based reagent | Forward primer of βactin RT-PCR | Fisher Scientific | | 5'-ATCCACATCTGCTGGAAGG T-3' |
| Peptide, recombinant protein | Alt-R S.p. Cas9 Nuclease 3NLS | IDT | 1074182 | |
| Peptide, recombinant protein | Platinum SuperFi DNA Polymerase | Invitrogen | 12351–010 | |
| Chemical compound, drug | CK666, Tocris Bioscience | Fisher Scientific | 395050 | |

*Continued on next page*

*Appendix 1—key resources table continued*

| Reagent type (species) or resource | Designation | Source or reference | Identifiers | Additional information |
|---|---|---|---|---|
| Chemical compound, drug | Alexa Fluor 488 Phalloidin | Thermo Fisher Scientific | A12379 | |
| Commercial assay or kit | HiScribe T7 High Yield RNA Synthesis Kit | New England Biolabs | E2040S | |
| Commercial assay or kit | RNeasy Mini Kit | QIAGEN | 74104 | |
| Software, algorithm | Zen2.1(Blue edition) | Carl Zeiss Microscopy | | http://www.zeiss.com |
| Software, algorithm | FIJI/ImageJ | *Schindelin et al., 2012* | | https://fiji.sc/ |
| Software, algorithm | RStudio | R Foundation for Statistical Computing | | https://www.r-project.org/ |

