## [Decision Letter]

**Acceptance summary:**

In the manuscript by Inaba et al. the authors investigate the role of intermediate filaments (Ifs) in the formation and growth of actin-based microridges in epithelial cells of the zebrafish epidermis. They identify a role for a complex of actin-interacting plakins evoplakin (Evpl) and periplakin (Ppl) in the initiation of microridges and their link to keratin in their elongation. The authors present beautiful imaging data along with a compelling combination of genetics and cell biology to dissect the molecular processes underlying microridge morphogenesis. This work provides insights into a new role for intermediate filaments and plakin family cytolinker proteins in cellular morphogenesis.

**Decision letter after peer review:**

Thank you for submitting your article "Keratins and Plakin family cytolinker proteins control the length of epithelial microridge protrusions" for consideration by *eLife*. Your article has been reviewed by four peer reviewers, including Michel Bagnat as the Reviewing Editor and Reviewer #1, and the evaluation has been overseen by Anna Akhmanova as the Senior Editor. The following individual involved in review of your submission has agreed to reveal their identity: Chen-Hui Chen (Reviewer #4).

The reviewers have discussed the reviews with one another and the Reviewing Editor has drafted this decision to help you prepare a revised submission.

We would like to draw your attention to changes in our revision policy that we have made in response to COVID-19 (https://elifesciences.org/articles/57162). Specifically, we are asking editors to accept without delay manuscripts, like yours, that they judge can stand as *eLife* papers without additional new experiments, even if they feel that they would make the manuscript stronger. Thus, the revisions requested below only address clarity and presentation, and minor addition of data that we hope you might already have at hand.

Summary:

In the manuscript by Inaba et al. the authors investigate the role of intermediate filaments (Ifs) in the formation and growth of actin-based microridges in epithelial cells of the zebrafish epidermis. The identify a role for a complex of actin-interacting plakins evoplakin (Evpl) and periplakin (Ppl) in the initiation of microridges and their link to keratin in their elongation.

The authors present beautiful imaging data along with a compelling combination of genetics and cell biology to dissect the molecular processes underlying microridge morphogenesis. This work provides insight into a valuable new role for intermediate filaments and plakin family cytolinker proteins in cellular morphogenesis. The manuscript is well written and is easy to read.

Overall, the data supports well their conclusions and the beautiful imaging presented illustrate well their work. There are however some missing pieces of support data and areas were additional data and/or editing of the text is needed.

Essential revisions:

1) The authors report new BAC transgenics for Evpl and Ppl that are used to characterize the role of these proteins in microridge morphogenesis.

1A) What is their pattern of expression? Whole mount pictures of live embryos and larvae and, if possible, cross sections of larvae should be provided as supplemental info.

1B) Are these BAC transgenes functional? i.e. do they rescue the mutants.

2) New mutant alleles for *evpl* and *ppl* are reported, but more basic info needs to be added.

2A) Please provide whole mount pictures of Mut/WT larvae, even if the phenotype is not obvious. Those can be in supplement and would be valuable for ZFIN.

2B) What happens to the transcripts? RT data is needed.

3) In Figure 3—figure supplement 2 the authors show images that suggest epidermal cells may show size and/or morphological differences compared to WT, please check if this is indeed the case and add results as part of the figure.

4) Dots in a dot plot should reflect independent replicates (e.g., different animals). One could argue that different cells from the same animal are not completely independent replicates. So, dependent (cells from the same animal) and independent samples (cells from different animals) are better not thrown together for graphing and statistical testing. Alternatively, you may take the averages from each animal, dot those, and test them against each other between samples.

5) Does latrunculin has the same effect as CK666. A second actin inhibitor could strengthen the drug results.

6) Subsection “Plakin-keratin interactions stabilize microridges”, related to Figure 7: the authors compare double mutant animals expressing full-length Evpl and Ppl to those expressing Evpl and Ppl head-rod domains, "to test if Plakin-keratin binding plays a role in stabilization." While the videos are convincing as to the differences in dynamics, can the authors look directly at keratin localization (perhaps using the Krt17 BAC transgenic?) to determine whether the loss of the Plakin tail domains did indeed affect keratin localization in microridges? This would be a useful control.

---

## [Author Response]

Essential revisions:1) The authors report new BAC transgenics for Evpl and Ppl that are used to characterize the role of these proteins in microridge morphogenesis.1A) What is their pattern of expression? Whole mount pictures of live embryos and larvae and, if possible, cross sections of larvae should be provided as supplementary information.

We now clarify in the revised manuscript that we detected expression of these reporters exclusively in periderm cells, as expected from RNA-seq data. We have not made stable transgenic lines with these reporters--all the images of them in the manuscript are of transient transgenic animals. Since transient expression of BAC reporters is highly mosaic, whole mount pictures would not provide an accurate representation of their expression patterns.

1B) Are these BAC transgenes functional? i.e. do they rescue the mutants.

Yes--we now show that the BAC reporters rescue microridge development, which is evident from comparing BAC-expressing cells in mutants to neighboring non-BAC-expressing cells (Figure 4—figure supplement 4). We have not quantified this effect, but have extensively quantified rescue with tagged cDNAs elsewhere in the manuscript.

2) New mutant alleles for evpl and ppl are reported, but more basic info needs to be added.2A) Please provide whole mount pictures of Mut/WT larvae, even if the phenotype is not obvious. Those can be in supplement and would be valuable for ZFIN.

We did not note any differences between the morphologies of mutant and WT fish. We have added whole mount pictures of mutant and wild-type larvae and adults (Figure 4—figure supplement 7).

Although not requested by reviewers, in the spirit of including data that may be useful to other zebrafish researchers, we have also added morpholino data showing that morpholino knockdown phenocopies the genetic mutants (Figure 4—figure supplement 5).

2B) What happens to the transcripts? RT data is needed.

We have added RT-PCR data confirming that the genomic deletions are indeed reflected in the mutant transcripts (Figure 4—figure supplement 1). Although we have not conducted rigorous quantitative RT-PCR, PCR products from mutants were notably weaker than from controls, indicating that the aberrant transcripts were likely destabilized by nonsense-mediated decay.

3) In Figure 3—figure supplement 2 the authors show images that suggest epidermal cells may show size and/or morphological differences compared to WT, please check if this is indeed the case and add results as part of the figure.

The supplementary figure noted by the reviewer was of epidermis covering adult scales. Since cells vary markedly in size and shape across the adult epidermis, determining if cell sizes or shapes are affected in mutant adults would require a broad survey of cells, which we felt could distract from the focus of the manuscript. However, to address the reviewer’s concern, we measured apical periderm cell areas in larvae (Figure 4—figure supplement 2C). The distribution of cell areas was indistinguishable between WT, *evpl-*/-, and *ppl-*/- animals, indicating that defects in microridge morphogenesis in these mutants are not a consequence of broader defects in cell morphology. Interestingly, we found that cells in *evpl-/-,ppl-/-* double mutants were significantly smaller than in the other genotypes. Since this cell size defect does not correlate with microridge morphology, this phenotype likely reflects a distinct, redundant role for these proteins in inhibiting apical constriction. This observation is now noted in the manuscript.

4) Dots in a dot plot should reflect independent replicates (e.g., different animals). One could argue that different cells from the same animal are not completely independent replicates. So, dependent (cells from the same animal) and independent samples (cells from different animals) are better not thrown together for graphing and statistical testing. Alternatively, you may take the averages from each animal, dot those, and test them against each other between samples.

As suggested by the reviewer, we averaged the cell microridge length averages for each animal and reconducted the analysis of microridge lengths in different genotypes (Figure 4B). This analysis yielded similar results to our initial analysis, which treated each cell separately. The only exception was that differences between *ppl+/-* and WT or *evpl+/-* animals were no longer significant, likely because of the substantially reduced sample size. The p-values for both analyses are now reported in Figure 4—figure supplement 3A-B. Since cell-to-cell variation in microridge length was greater than animal-to-animal variation (suggesting that microridge length is determined cell-autonomously), plotting the average of the averages would minimize the range of variation, so continue displaying the data from all cells, but, to allow readers to appreciate animal-to-animal variation in cell averages, we have also provided an alternative version of the plot of average microridge length/cell in each genotype, with color coded data points to indicate which cells were from the same animals (Figure 4—figure supplement 2A).

Finally, to provide an alternative analysis of these data similar to analyses in previous studies of microridges (Raman et al., 2016; Magre et al., 2019; van Loon et al., 2020), we now also show the distribution of the lengths of all protrusions in all cells pooled as violin plots, with accompanying statistical analyses (Figure 4—figure supplement 2B, and Figure 4—figure supplement 3C).

5) Does latrunculin has the same effect as CK666. A second actin inhibitor could strengthen the drug results.

Latrunculin indeed causes microridges to disassemble, but it also disassembles pegs, disrupts cell-cell junctions, makes cells and animals sick, and causes significant cell death in the skin. For these reasons it would not be an appropriate tool for these experiments. By contrast, the CK666 treatment regime we used did not notably affect cell-cell junctions or cell viability.

6) Subsection “Plakin-keratin interactions stabilize microridges”, related to Figure 7: the authors compare double mutant animals expressing full-length Evpl and Ppl to those expressing Evpl and Ppl head-rod domains, "to test if Plakin-keratin binding plays a role in stabilization." While the videos are convincing as to the differences in dynamics, can the authors look directly at keratin localization (perhaps using the Krt17 BAC transgenic?) to determine whether the loss of the Plakin tail domains did indeed affect keratin localization in microridges? This would be a useful control.

This was an excellent suggestion. Unfortunately, although we made considerable efforts to carry out this experiment, we were unable to overcome technical challenges. To address this question we injected two separate plasmids encoding the head-rod domains of Evpl and Ppl, along with a Krt17 BAC reporter, into double mutants, and needed to identify cells expressing all three constructs simultaneously. Unfortunately, we were not able to achieve rescue with tripartite expression, even in controls expressing full-length Evpl and Ppl. This experiment should be possible when we have transgenic lines, which we plan to create for future studies. For now, we realize that our interpretation (i.e. protrusions in cells expressing the head-rod domains are less stable due to a deficit in keratin recruitment) is based on an assumption that these protrusions lack keratin. Although the suggested experiment would have been the definitive way to test this assumption, we believe that the rest of the data in our paper, and previous biochemical work with these proteins, make this a reasonable assumption.